# The Interannual Changes in the Secondary Production and Mortality Rate of Main Copepod Species in the Gulf of Gdańsk (The Southern Baltic Sea)

**Lidia Dzierzbicka-Głowacka** [1,*]👤, **Maja Musialik-Koszarowska** [1], **Marcin Kalarus** [2], **Anna Lemieszek** [2], **Paula Prątnicka** [3], **Maciej Janecki** [1]👤 **and Maria Iwona Żmijewska** [3]

[1] Institute of Oceanology, Polish Academy of Sciences, Powstańców Warszawy 55, 81-712 Sopot, Poland; m.m.koszarowska@gmail.com (M.M.-K.); mjanecki@iopan.pl (M.J.)

[2] Maritime Institute of Gdańsk, Długi Targ 41/42, 80-830 Gdańsk, Poland; Marcin.Kalarus@im.gda.pl (M.K.); Anna.Lemieszek@im.gda.pl (A.L.)

[3] Institute of Oceanography, University of Gdańsk, Av. Marszałka Piłsudskiego 46, 81-378 Gdynia, Poland; paula.kacprzak@phdstud.ug.edu.pl (P.P.); maria.iwona.zmijewska@ug.edu.pl (M.I.Ż.)

\* Correspondence: dzierzb@iopan.pl; Tel.: +48-58-731-1915

**Abstract:** The main objective of this paper was description of seasonal and interannual trends in secondary production and mortality rates of the three most important Copepoda taxa in the Gulf of Gdańsk (southern Baltic Sea). Samples were collected monthly from six stations located in the western part of the Gulf of Gdańsk during three research periods: 1998–2000, 2006–2007, and 2010–2012. Production was calculated based on copepod biomass and mortality rates estimated according to vertical life table approach. Redundancy analysis was used to investigate relationship between secondary production and environmental conditions. During the entire research period there was significant interannual and seasonal variability of secondary production, mortality rate, as well as abundance and biomass anomalies. Conducted analysis revealed positive correlation between increasing temperature and production of *Acartia* spp. and *Temora longicornis* developmental stages, while older copepodites of *Pseudocalanus acuspes* showed almost negative correlation with temperature. The mortality rate estimations obtained for *Acartia* spp. were the highest in summer, while *Temora longicornis* peaked in spring–summer period. The lowest mortality rate estimations were noted in autumn and winter for almost all stages of investigated taxa.

**Keywords:** Copepoda; secondary production; mortality rates; Baltic Sea; Gulf of Gdańsk

## 1. Introduction

The Baltic Sea is a unique ecosystem, and due to its inland character, large drainage area, and limited exchange of sea water with the Atlantic it is very sensitive to ongoing natural and anthropogenic (climate change, pollution, eutrophication, and overfishing) changes. The coastal zone is especially vulnerable, and in similarity to other regions of the Baltic Sea, exhibits little variety in the number of animal species, which is the result of eutrophication and the degradation of the environment. Despite this, it is considered among the marine habitats with the highest biological productivity. It plays an important ecological role by offering a variety of habitat types for many species, giving shelter to animals, and functioning as nursery areas and feeding grounds for many marine fishes and crustaceans.

In marine pelagic food webs, zooplankton plays a key role as an important link in energy transfer between primary producers and higher tier consumers, strongly influencing fish production. Zooplankton of the Gulf of Gdańsk typically consist of euryhaline and eurythermic taxa, among

copepods mainly species from genera Acartia and *Temora longicornis*, as well as the less abundant, but ecologically important, *Pseudocalanus acuspes*. They are preferred prey items for commercially important fishes like sprat and herring as well as larval cod.

In order to properly asses the role of zooplankton in the marine food web, zooplankton secondary production and mortality rates need to be estimated. It is a useful tool to obtain knowledge of marine productivity, quantifying, behavior, distribution, migration patterns, and transfers between food web components [1–3]. Many studies have focused on vital rates of copepods in the laboratory [4–10], yet only few studies exist on its population dynamics in the field [11–17].

The aim of this study was to describe the secondary production and mortality rates of three dominant calanoid copepod species—*Acartia* spp., *Temora longicornis*, and *Pseudocalanus acuspes*—in the southern Baltic Sea with relation to hydrographic water conditions. The data will be used for upgrading the copepod population model for the Baltic Sea [10,16,18–20].

## 2. Materials and Methods

### 2.1. Study Area

The Gulf of Gdańsk is a widely open gulf located in the southern part of the Baltic Sea between Poland and Russia. The western part of the bay can be separated into a shallow part known as Puck Bay and further to the west the semi-enclosed Puck Lagoon. The bay is strongly influenced by river waters, especially by its largest river, the Vistula, which on average brings 1080 m$^3$ s$^{-1}$ of fresh water. The average depth of the gulf is ~50 m, with a maximum depth (Gdańsk Deep) of 118 m. Water temperature ranges from over 20 °C at the surface during summer, with the maximum usually in August, to ~2 °C in February. Water stratification is frequent during the warmer months, leading to the occurrence of seasonal thermocline, while during the winter gulf waters become well-mixed. Due to the brackish character of the Baltic Sea, gulf water salinity stays within the range of 7 to 8. Surface waters, especially in the coastal region, can be less saline due to river discharge. Halocline is present in the deepest part of the gulf, mainly in the region of Gdańsk Deep. Due to high eutrophication and frequent algal bloom, water transparency varies highly depending on the season, from a few meters up to 16 m.

Hydrology of the gulf is heavily impacted by the Vistula River, which is the largest river flowing into the bay, bringing fresh water and nutrients. The gulf is also the location of the largest Polish ports—Gdańsk and Gdynia—which have a significant impact on its environment due to pollution, sea transport, and fishing.

### 2.2. Sampling

Sampling stations were located at a depth gradient in the central part of the bay (stations 1–5); one station was also located in the shallower, semi-enclosed Puck Bay (6) (Figure 1).

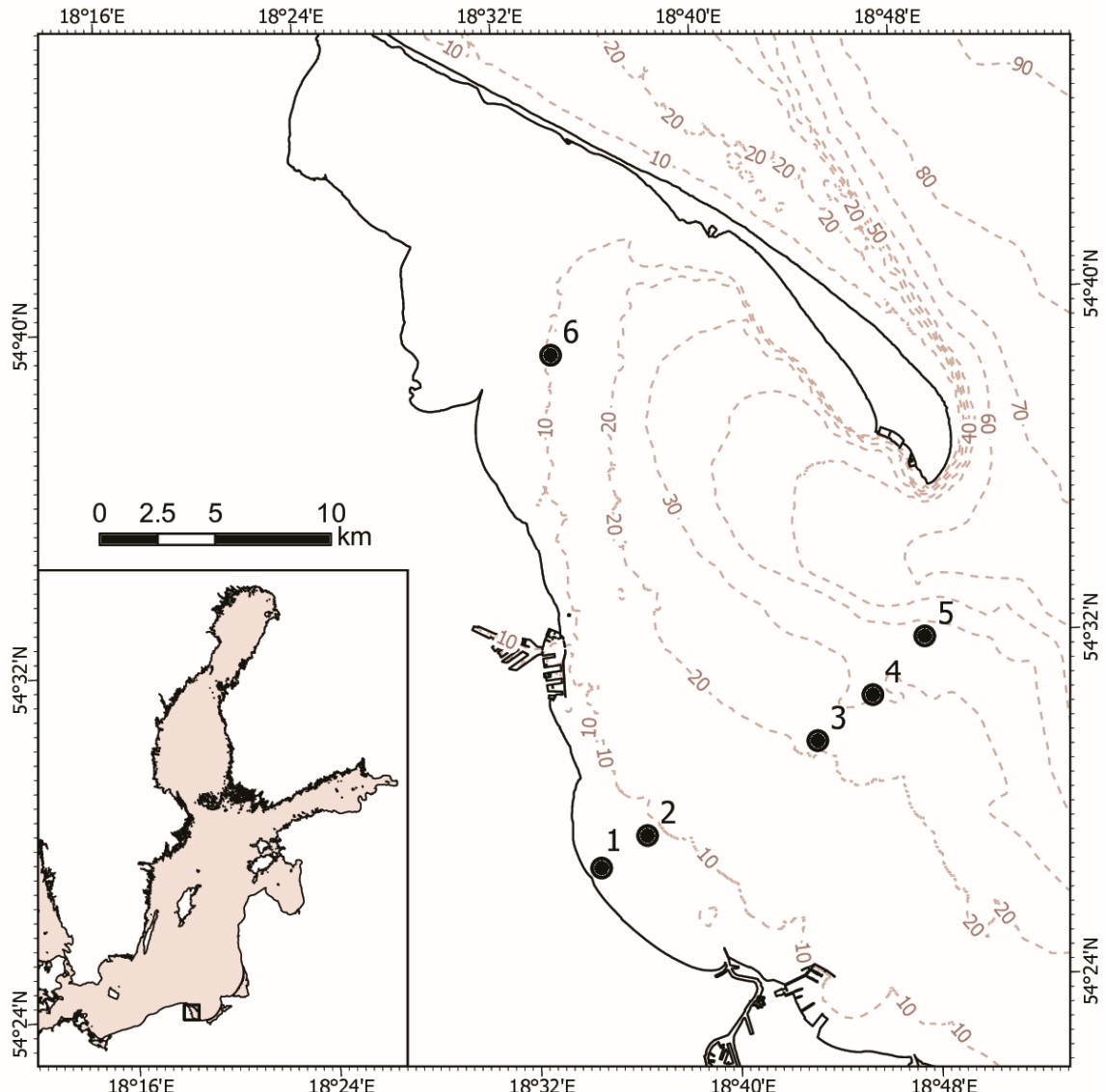

**Figure 1.** Study area and location of the sampling stations in the Gulf of Gdańsk (southern Baltic Sea).

Conducted research included investigation of spatial and temporal variations of hydrological conditions (temperature and salinity) as well as investigations of the qualitative and quantitative structure of three main copepod taxa (*Acartia* spp., *Temora longicornis*, and *Pseudocalanus acuspes*). Sampling was conducted during three separate projects spanning almost 14 years from 1998 to 2012. The first sampling period lasted from August 1998 to September 2000, the second took place from 2006 to 2007, and the last one from 2010 to 2012. During the sampling campaigns zooplankton samples were collected with almost monthly coverage. Two types of sampling nets were used: from 1998 to 2007 samples were collected with a Copenhagen type net (Institute of Oceanography, University of Gdańsk, Gdynia, Poland) and, since 2010, the WP-2 type net (Institute of Oceanography, University of Gdańsk, Gdynia, Poland) was utilized. Both types of nets were of 100-μm mesh size. Samples were collected with vertical hauls; at shallower stations (1, 2, and 6) haul was conducted from the bottom to the surface, while at deeper stations (3, 4, and 5) the water column was divided into 10-m layers. All samples were collected during the daytime (mainly between 11 am and 2 pm) so the diurnal vertical migrations were not accounted for. Along with the collection of biological material water, physicochemical conditions (T, S) were measured (Cond 3110, WTW Xylem Analytics, Germany).

## 2.3. Model Data

The source of the numerical results used in the manuscript is the "Baltic Sea Biogeochemical Reanalysis" product which provides a 24-year biogeochemical reanalysis for the Baltic Sea (1993–2016) using the ice–ocean model NEMO-Nordic (based on NEMO-3.6, Nucleus for European Modelling of the Ocean) coupled with the biogeochemical model SCOBI (Swedish Coastal and Ocean Biogeochemical Model), together with LSEIK data assimilation. Values for all the presented variables (dissolved oxygen and chlorophyll *a*) have been derived from the daily means. Direct files have been accessed and downloaded upon registration via a Copernicus Marine Environment Monitoring Service's (CMEMS) and choosing the BALTICSEA_REANALYSIS_BIO_003_012 product. Detailed description of the product is available as an online resource in the documentation section. In particular, a product user manual [21] that covers a detailed production subsystem description as well as a quality report of the product [21], which provides an estimated accuracy numbers for each variable summarized in the table below (Table 1).

**Table 1.** Estimated accuracy of the numerical model data according to quality information document.

| RMS Error | 0–5 m | 5–30 m | 30–80 m |
|---|---|---|---|
| Dissolved oxygen (mmol m$^{-3}$) | 19 | 38 | 51 |
| Chlorophyll a (mg m$^{-3}$) | 5 | 3 | 0.9 |

Each in situ station has been paired with the closest grid cell in the product's model domain estimating the minimal distance between stations; each grid cell uses longitude and latitude meta-information. Vertical means were calculated for the related timeseries.

## 2.4. Biomass and Abundance

Collected zooplankton samples were analyzed in terms of qualitative and quantitative descriptions of three copepod taxa, key for southern Baltic mesozooplankton populations: *Acartia* spp. (including *A. bifilosa*, *A. longiremis*, and *A. tonsa*), *Temora longicornis*, and *Pseudocalanus acuspes*. All analysis was performed in accordance with the HELCOM COMBINE methodology [22]. Obtained abundances were then used to calculate the numbers of individuals per m$^2$ and m$^3$. Finally, standard weights [23] were used to estimate the biomass values (mg C) of each of the taxa per m$^3$.

Obtained biological data were normalized by transforming to natural logarithms (ln(x+1)). Copepod abundances and biomasses were averaged over stations and the seasonal cycle was removed by subtracting long-term monthly means from annual monthly means.

Abundance values (ind. m$^{-2}$) were next used to calculate secondary production rates and mortality rates of the above-mentioned copepod taxa.

## 2.5. Secondary Production

Production rates of the investigated species were calculated with the Edmondson and Winberg equation [24]. Calculations were carried out for each of the copepodite stages with assumption of nonlimiting food conditions:

$$PC_i = N_i \times \Delta W_i / D_{\min_i} \tag{1}$$

where $PC_i$ is daily potential production of stage $i$ (wet weight), $i$ is the development stage, $D_{\min_i}$ is the development time of stage $i$ (day$^{-1}$), $\Delta W_i$ is the weight increase of stage $i$, and $N_i$ is the abundance of stage $i$ (ind. m$^{-2}$). $D_{min}$ of developmental stages was computed using the function provided by Figiela et al. [25]:

$$D_{min} = f(T) \tag{2}$$

where $D_{min}$ is the minimum value of the development time, for which the growth rate of an individual is not limited by food availability. The wet weights of the copepodite stages and adults were accepted

after Hernroth [23]. The conversion factor of 0.05 after Mullin [26] was used to transform the wet weight to carbon content.

The development time $D$ is a function of three variables: concentration of food *Food* (food availability varying depending on season [16]), temperature $T$, and salinity $S$. $D_{min}$ was described for each species at the nauplius and I–V copepodite developmental stages by the equations described by Figiela [25].

For *Acartia* spp.:

$$ft = \begin{cases} 1, \ T \leq 19\,°C \\ 0.9957e^{0.0187(T-19)}, \ T \geq 19\,°C \end{cases} \tag{3}$$

$$fs = 2 - (1 - \exp(-0.9(S - 0.001))) \tag{4}$$

*Acartia* spp. nauplii:

$$D_N = \left(31.34e^{-0.092\cdot T} + 4921.7\,Food^{-1.7462}e^{-(0.1805\,Food^{-0.1061})\cdot T}\right)ft\cdot fs \tag{5}$$

*Acartia* spp. copepodite:

$$D_C = \left(40.956e^{-0.0849\cdot T} + 1178.5\,Food^{-1.0486}\,e^{-(0.0739\,Food^{0.1059})\cdot T}\right)ft\cdot fs \tag{6}$$

For *Temora longicornis*:

$$ft = \begin{cases} 1, \ T \leq 15\,°C \\ 0.9972e^{0.0269(T-15)}, \ T \geq 15\,°C \end{cases} \tag{7}$$

$$fs = 2 - (1 - \exp(-0.5(S - 2))) \tag{8}$$

*Temora longicornis* nauplii:

$$D_N = \left(39.565e^{-0.0964\cdot T} + 61\,e^{-0.0081\,Food}e^{-(0.0006\,Food+0.0588)\cdot T}\right)ft\cdot fs \tag{9}$$

*Temora longicornis* copepodite:

$$D_C = \left(38.693e^{-0.0809\cdot T} + 57.438\,Food^{-0.0037}\,e^{-(0.0007\,Food+0.0517)\cdot T}\right)ft\cdot fs \tag{10}$$

For *Pseudocalanus acuspes*:

$$ft = \begin{cases} 1, \ T \leq 14\,°C \\ 0.9993e^{0.0377(T-14)}, \ T \geq 14\,°C \end{cases} \tag{11}$$

$$fs = 3 - (2 - \exp(-0.25(S - 9))) \tag{12}$$

*Pseudocalanus acuspes* nauplii:

$$D_N = \left(41.342e^{-0.069\cdot T} + 2.679\,T^{1.0988}\,e^{(0.0209\ln T-0.0829)\cdot Food}\right)ft\cdot fs \tag{13}$$

*Pseudocalanus acuspes* copepodite:

$$D_C = \left(34.888e^{-0.0781\cdot T} + 1.786\,e^{1.0988\ln T}\,e^{(-0.0559e^{-0.0486\,T})\cdot Food}\right)ft\cdot fs \tag{14}$$

## 2.6. Mortality Rate

Mortality rates of the three investigated taxa were computed with the Aksnes and Ohman [1] method, which is based on the abundances at different developmental stages.

While estimating mortality rate of stage $i$ and $i + 1(\theta)$ stage duration was considered for a period equal to the corresponding duration of two consecutive stages ($\alpha_i + \alpha_{i+1}$), and it is assumed that the

two successive stages are taken impartially and are under the same influence of transport processes during these stages. This led to the following formula of mortality estimates [1].

For nauplii and copepodite stages CI–CIV:

$$\frac{e^{mD_i} - 1}{1 - e^{-mD_{i+1}}} = \frac{Z_i}{Z_{i+1}} \tag{15}$$

For copepodite stage CV (since the stage duration of adults is infinite):

$$m = \frac{(\ln \frac{Z_i}{Z_{adult}}) + 1}{D_i} \tag{16}$$

where $Z_i$ is the abundance of the developmental stage $i$, $Z_{adult}$ is the abundance of adults, $Z_{i+1}$ is the abundance of next stage ($i + 1$), $m_i$ is instantaneous mortality rate of stage $i$ ($d^{-1}$), and $D$ followed by subscripts are stage durations of copepodite stages $i$ and $i + 1$.

*2.7. Statistical Analyses*

The obtained biomass of copepods developmental stages was square root transformed prior to analysis. Similarities between samples were examined using the Euclidean distance index, depicted as a nonmetric multidimensional scaling (nMDS) [27], which illustrated similarities between analyzed seasons. One-way Analyses of Similarities (ANOSIM) were performed in order to test the significance between sampling seasons, years and stations. The association between temperature and each developmental stage of all investigated copepod species was fitted using generalized linear models (GLMs) with normal distribution. Additionally, means plots were carried out to illustrate the production and mortality of different developmental stages of copepods according to analyzed seasons. All analyses were performed in PRIMER (v 7, PRIMER-e, Massey University, Albany Auckland, New Zealand) [27] and CANOCO (v 5, Microcomputer Power, NY, USA) [28].

## 3. Results

*3.1. Hydrology*

The water temperature was characterized by a very similar distribution throughout the whole study period, with summer-to-winter fluctuations typical for the temperate climate region. Mean values of water temperature noted at the investigated region ranged from 0.57 ± 0.43 °C in March 2006 to 18.42 ± 3.03 °C in August 2010 (Figure 2).

Due to technical reasons, salinity values were not recorded during the first sampling campaign. Recorded salinity fluctuations were minimal, and did not exceed 1; mean values ranged from 6.32±0.50 in July 2010 to 7.60±0.09 in March 2012 (Figure 2).

The mean oxygen concentration during the study was also relatively constant. The maximum values were observed in the spring (~400 mmol $O_2 \cdot m^{-3}$) and the minimum in August/September (~300 mmol $O_2 \cdot m^{-3}$) (Figure 2). The oxygen concentration was at the same level at all sampling stations except stations shallower than 10 m (1, 2, and 6), where it was generally slightly lower.

The highest mean values of chlorophyll *a* concentration was reached in May of 2010, 2011, and 2012, with an average value of ~9 mg m$^{-3}$. In the spring/summer period the values were generally over 5 mg m$^{-3}$. The minimum chlorophyll *a* concentration was always noted in the winter. In the annual cycle, usually two different peaks in chlorophyll *a* concentration are observed in May and September (Figure 2). The highest values of chlorophyll *a* concentration were noted at shallow stations (1, 2, and 6), and the lowest at the deepest sampling station (5).

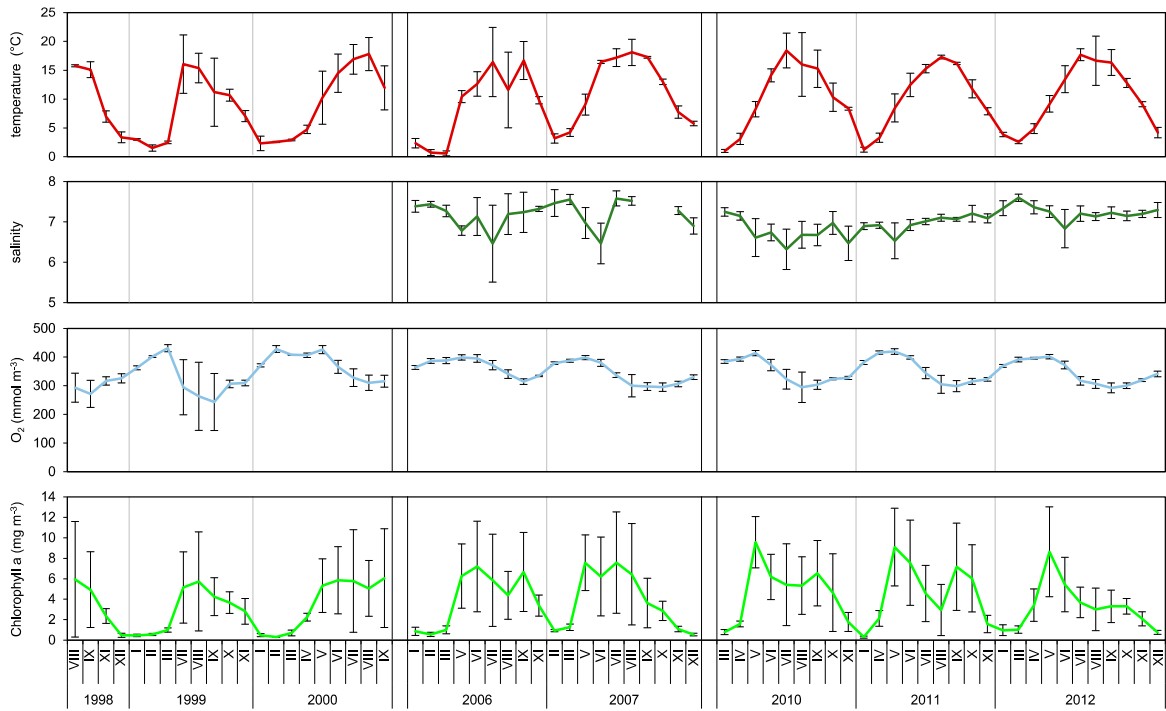

**Figure 2.** Water temperature, salinity, dissolved oxygen, and chlorophyll *a* (± standard deviation) in the Gulf of Gdańsk.

## 3.2. Abundance and Biomass Anomalies

Both the abundance and biomass of the considered taxa were highly seasonal. After a decrease at the beginning of the year, mainly in February, peak abundances and biomasses were found in summer. In the case of *Acartia* spp. highest abundance was observed in August while biomass peaked in July; for *T. longicornis*, both abundance and biomass peaks were found in July; while *P. acuspes* had it abundance peak in May, for biomass it was in August. The seasonal development, with maximum abundance and biomass in summer, is confirmed by monthly long-term means (Tables 2 and 3).

**Table 2.** Long-term monthly means (± standard deviation) for abundance (ln ind. m$^{-3}$) of three copepod taxa.

| Month | *Acartia* **spp.** | *T. longicornis* | *P. acuspes* |
|---|---|---|---|
| January | 7.28 ± 0.30 | 5.26 ± 0.63 | 4.27 ± 0.43 |
| February | 6.45 ± 0.63 | 4.62 ± 1.26 | 4.34 ± 1.60 |
| March | 6.74 ± 0.44 | 4.87 ± 0.62 | 5.21 ± 0.71 |
| April | 6.88 ± 1.15 | 5.09 ± 1.11 | 4.86 ± 0.83 |
| May | 7.85 ± 0.70 | 8.19 ± 1.04 | 4.95 ± 0.52 |
| June | 8.23 ± 0.56 | 8.49 ± 0.41 | 4.38 ± 0.76 |
| July | 9.41 ± 0.69 | 9.10 ± 0.87 | 4.04 ± 2.34 |
| August | 9.46 ± 0.75 | 7.58 ± 0.93 | 4.38 ± 2.83 |
| September | 9.30 ± 0.78 | 6.94 ± 1.14 | 3.02 ± 2.10 |
| October | 7.98 ± 0.27 | 7.41 ± 0.15 | 3.29 ± 0.92 |
| November | 7.18 ± 0.49 | 7.46 ± 0.66 | 4.04 ± 0.78 |
| December | 6.83 ± 0.83 | 6.39 ± 0.69 | 2.78 ± 2.42 |

**Table 3.** Long-term monthly means (± standard deviation) for biomass (ln mgC m$^{-3}$) of three copepod taxa.

| Month | *Acartia* spp. | *T. longicornis* | *P. acuspes* |
|---|---|---|---|
| January | 2.08 ± 0.21 | 1.54 ± 0.48 | 0.69 ± 0.24 |
| February | 1.39 ± 0.74 | 1.29 ± 0.98 | 1.01 ± 1.17 |
| March | 1.62 ± 0.47 | 1.26 ± 0.44 | 1.01 ± 0.52 |
| April | 1.67 ± 0.80 | 1.13 ± 0.53 | 0.67 ± 0.33 |
| May | 2.55 ± 0.68 | 2.94 ± 0.69 | 0.41 ± 0.20 |
| June | 2.79 ± 0.59 | 3.21 ± 0.38 | 0.29 ± 0.12 |
| July | 4.08 ± 0.64 | 3.85 ± 0.77 | 0.62 ± 0.85 |
| August | 3.91 ± 0.56 | 3.01 ± 1.13 | 1.07 ± 0.80 |
| September | 3.83 ± 0.80 | 1.92 ± 0.80 | 0.40 ± 0.49 |
| October | 2.69 ± 0.27 | 2.45 ± 0.21 | 0.25 ± 0.21 |
| November | 1.87 ± 0.40 | 2.50 ± 0.58 | 0.47 ± 0.30 |
| December | 1.40 ± 0.53 | 1.52 ± 0.53 | 0.30 ± 0.28 |

The abundance and biomass of *Acartia* spp. (Figure 3) showed negative nonseasonal anomalies during the first research period (1998–2000); it became positive during the second research period (2007) and at the beginning of the third period (2010) to later become mostly negative (2011–2012). The second of the investigated copepods, *T. longicornis*, also showed negative nonseasonal anomalies during the period from 1998 to 2000, while in the second research period (2006–2007) observed anomalies were mostly positive. This was also true for the beginning of 2010; later it came closer to 0, and again positive at the end of 2011 (Figure 4). Anomalies for *P. acuspes* showed a similar trend, with negative anomaly during the first research period, especially during 1998 and 1999. During the second research period observed anomalies for that species were mostly positive, with a drop at the end of 2007. In the third research period anomalies were mostly positive, especially in 2010, with occasional negative values in 2011 and 2012 (Figure 5).

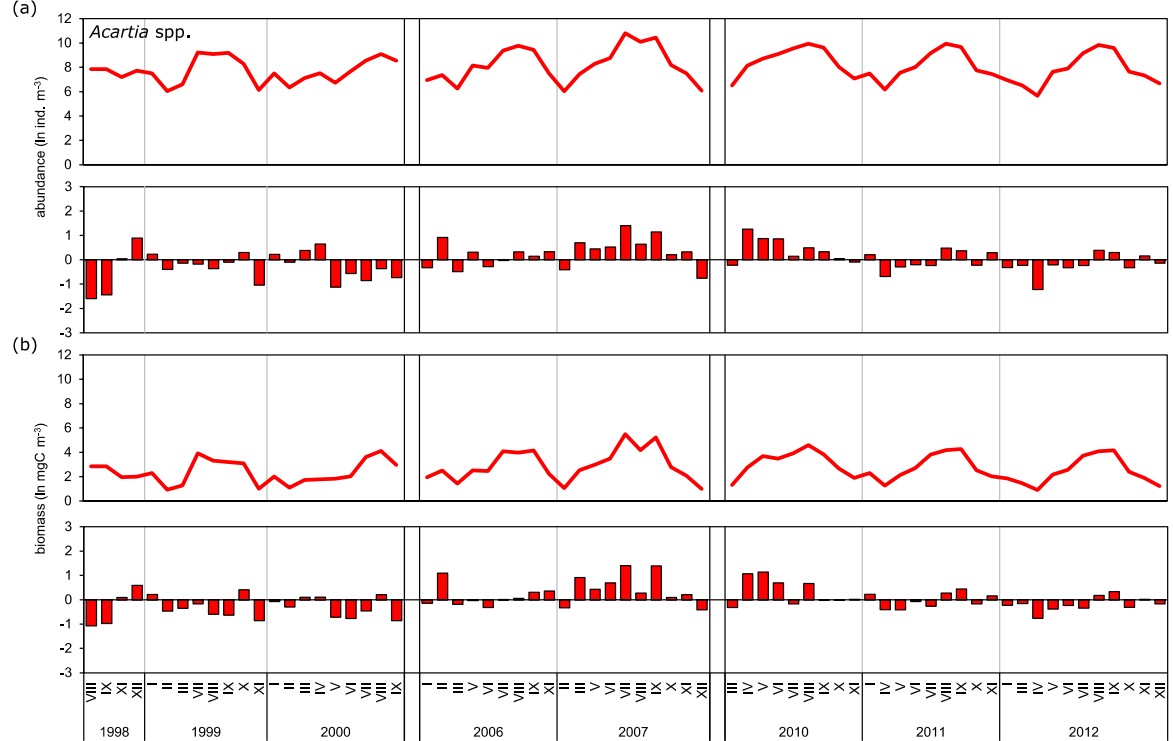

**Figure 3.** Interannual (**a**) abundance and (**b**) biomass monthly mean and anomaly of *Acartia* spp.

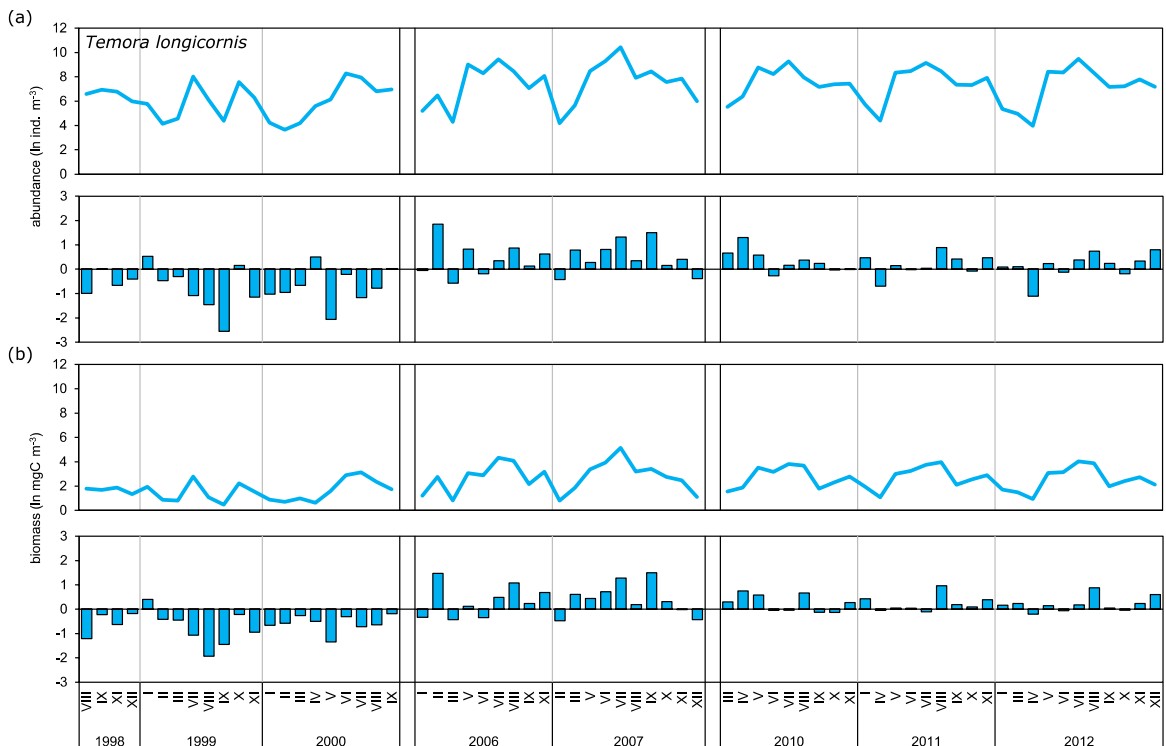

**Figure 4.** Interannual (**a**) abundance and (**b**) biomass monthly mean and anomaly of *Temora longicornis*.

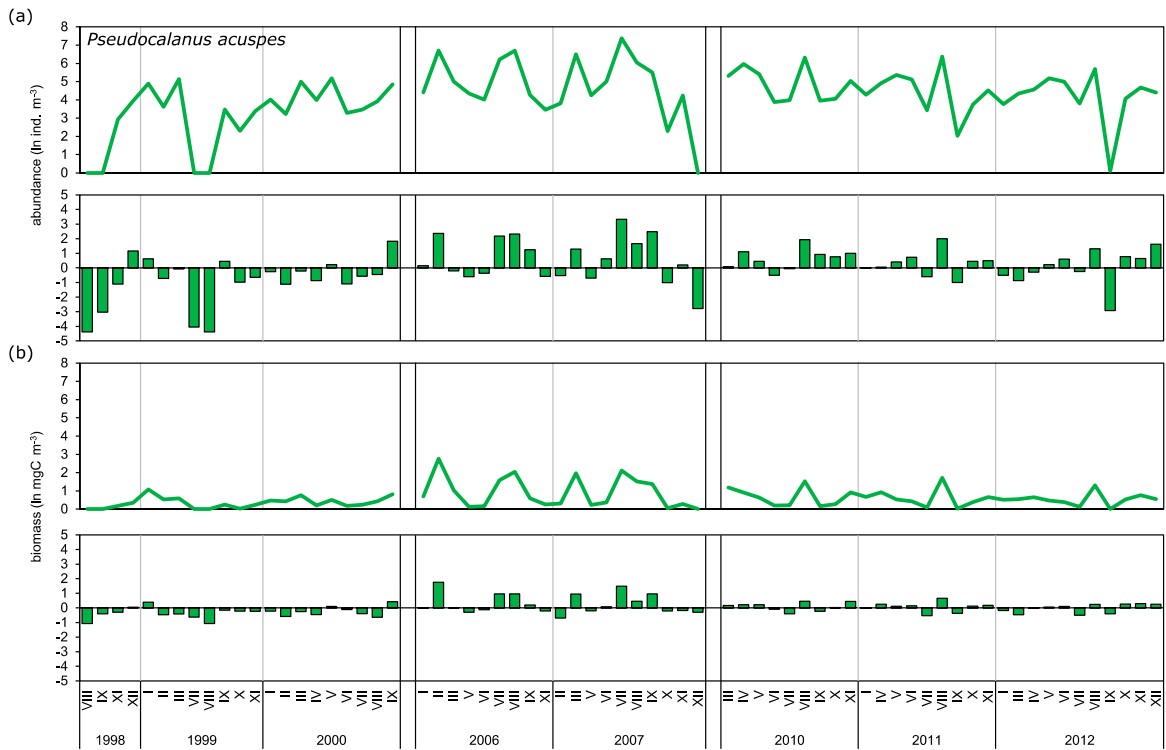

**Figure 5.** Interannual (**a**) abundance and (**b**) biomass monthly mean and anomaly of *Pseudocalanus acuspes*.

### 3.3. Secondary Production

Considering the entire research period, interannual and seasonal variability of secondary production of *Acartia* spp., *T. longicornis*, and *P. acuspes* was visible. Obtained results showed that the highest average secondary production was recorded during 2006 and 2007, while the lowest

values were observed from 1998 to 2000. During each of the years, the summer season was characterized by the highest values of production rates (Figure 6).

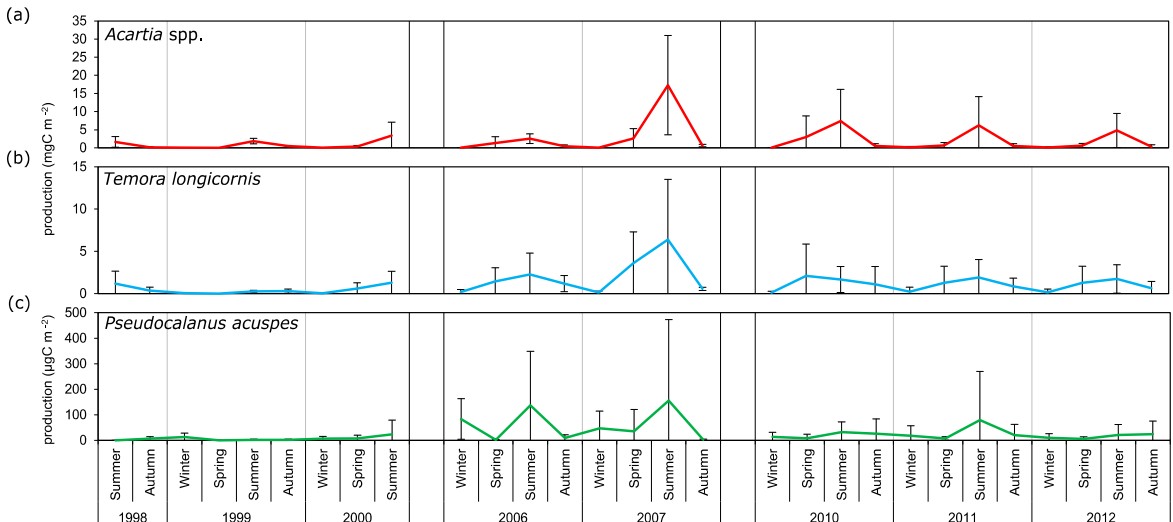

**Figure 6.** Mean secondary production rates (± standard deviation) of for (**a**) *Acartia* spp., (**b**) *Temora longicornis*, and (**c**) *Pseudocalanus acuspes* in the Gulf of Gdańsk during particular seasons.

During the first research period, the average production rate for *Acartia* spp. reached the highest value in the summer of 2000, almost 4 mg C m$^{-2}$. In the second study period (2006–2007), there was a large variation in the production rate. In summer 2007 it was more than four times higher (~16 mg C m$^{-2}$) than recorded in the previous year. In the last years of research, a downward tendency in production was observed, from ~8 mg C m$^{-2}$ in the summer of 2010, to ~6 mg C m$^{-2}$ the following year, and finally ~4 mg C m$^{-2}$ in summer 2012 (Figure 6).

The secondary production of *T. longicornis* was lower than for *Acartia* spp. In the first years of the study, from 1998 to 2000, the average maximum value in the summer fluctuated around 1 mg C m$^{-2}$. During the second time interval, similarly to *Acartia* spp., significant variations in production values were observed, reaching ~2 mg C m$^{-2}$ in summer 2006, and ~6 mg C m$^{-2}$ a year later. In following years of research, the highest average production rates were observed in spring–summer periods. In 2010 it fluctuated between 2 mg C m$^{-2}$ in spring to 1.8 mg C m$^{-2}$ in summer, while in 2011 and 2012 noted values were 1.7 mg C m$^{-2}$ in spring and 2 mg C m$^{-2}$ in summer (Figure 6).

Among the investigated copepods, *P. acuspes* was characterized by the lowest secondary production values. In the 1998 to 2000 period, the average values of production rates of the species did not exceed 25 µg C m$^{-2}$ (summer 2000). During the second period of our research, two production peaks were observed during each year. In 2006, the first was recorded in summer with an average value of ~175 µg C m$^{-2}$, and the second in autumn—~80 µg C m$^{-2}$. In 2007 these values were similar: in the summer the average production was 150 µg C m$^{-2}$ and in autumn it reached approximately 50 µg C m$^{-2}$. During the last research period even lower production rate values were observed. In 2010 and 2012, the average values did not exceed 50 µg C m$^{-2}$, and the maximum was observed in the summer of 2011, reaching 70 µg C m$^{-2}$ (Figure 6).

Obtained production values show significant differences between each designated season (winter, spring, summer, and autumn) (ANOSIM, $p = 0.001$, global R = 0.454), which is quite clearly visible on the nMDS plot (Figure 7). There is, however, a visible connection between spring and autumn groups, which overlap partially ($p = 0.001$, global R = 0.089). The largest differences in production rates were recorded between summer and winter ($p = 0.01$, global R = 0.841).

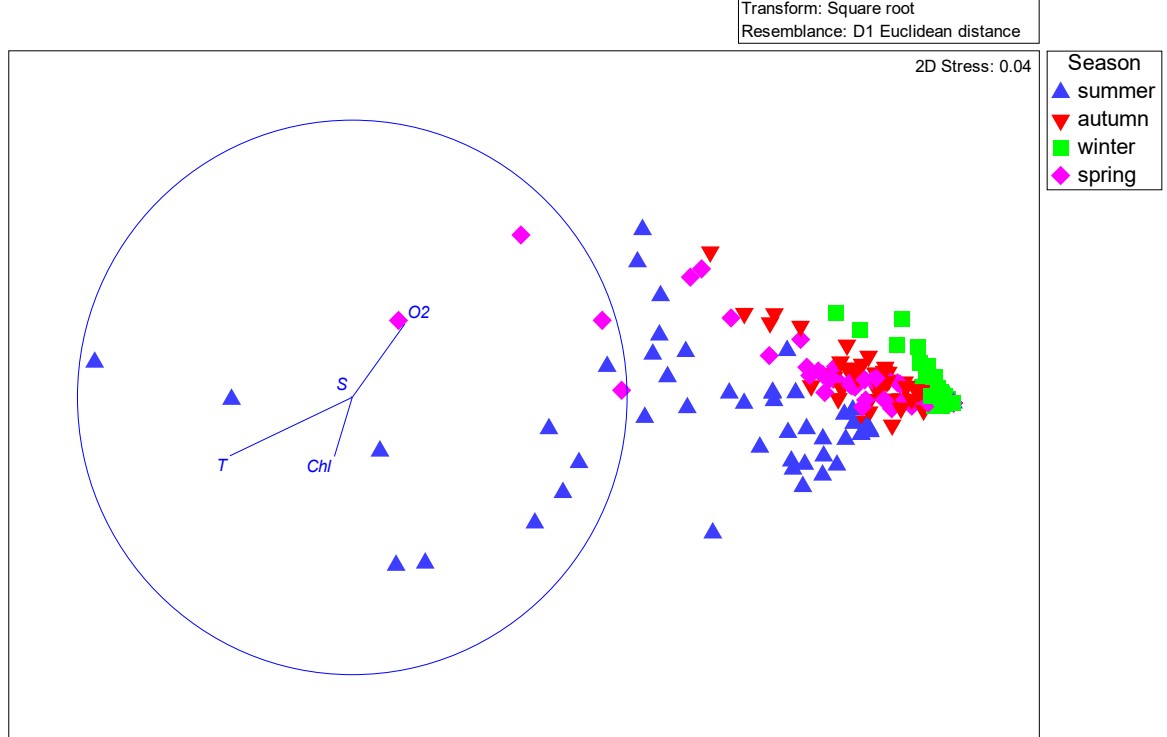

**Figure 7.** Nonmetric multidimensional scaling (nMDS) plot illustrating the samples ordination according to production rates based on the seasons factor.

Generally, production rates of *Acartia* spp. for all stages were the highest in summer season (Figure 8). Also, the range of production rates was highest in case of this taxon. *T. longicornis* showed a similar trend, but during spring, production values were even higher than in summer, especially for the youngest copepodites (CI-CIII). What is more, *T. longicornis* production values were generally lower than for *Acartia* spp. (Figure 8). There was no visible tendency for production distribution of *P. acuspes* stages. Nauplii production was the lowest in autumn, but during all seasons there were very high discrepancies in production ranges. The production values of copepodites CII and CIII were the highest during the summer season, while the production rates of older CIV and CV increased in winter (Figure 8).

GLM analysis revealed that with increasing temperature, the production of *Acartia* spp. and *T. longicornis* developmental stages increased, while stage CV of *P. acuspes* indicated almost neutral correlation with temperature. The most intensive production rise with increasing temperature demonstrated *Acartia* spp. stages (Table 4, Figure 9).

Horizontal distribution showed that the lowest average production rates of the three species in the Gulf of Gdańsk were recorded at the shallow stations (1, 2, and 6), while the deeper stations, mainly 4 and 5, were characterized by the highest production. The difference between production values between the stations was particularly visible in the spring and summer seasons. Horizontal distribution of secondary production of *Acartia* spp. showed the highest values for all six stations in the summer (Figure 10). In the case of *T. longicornis* the highest production was observed in spring at station 5, while in the summer the maximum production was recorded at station 4 (Figure 11). For *P. acuspes*, the highest production was noted in summer at station 5, while the lowest values were observed at shallow stations 1 and 6 thorough all seasons (Figure 12).

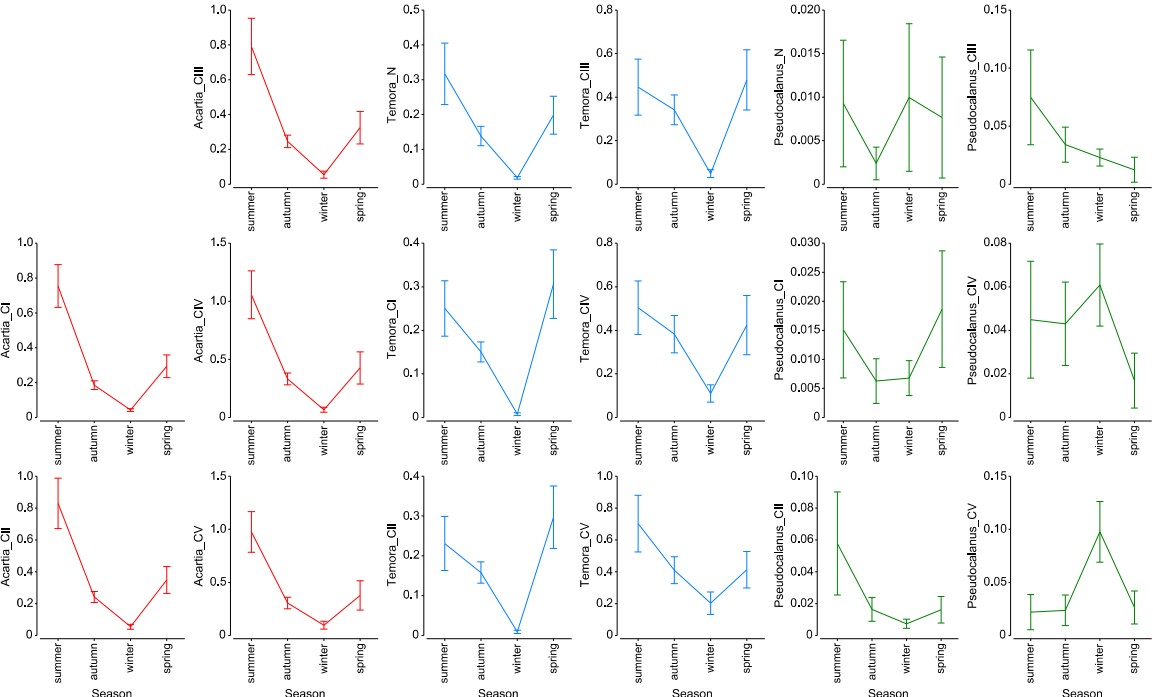

**Figure 8.** Means plot illustrating production distribution of *Acartia* spp., *Temora longicornis*, and *Pseudocalanus acuspes* developmental stages in relation to seasons.

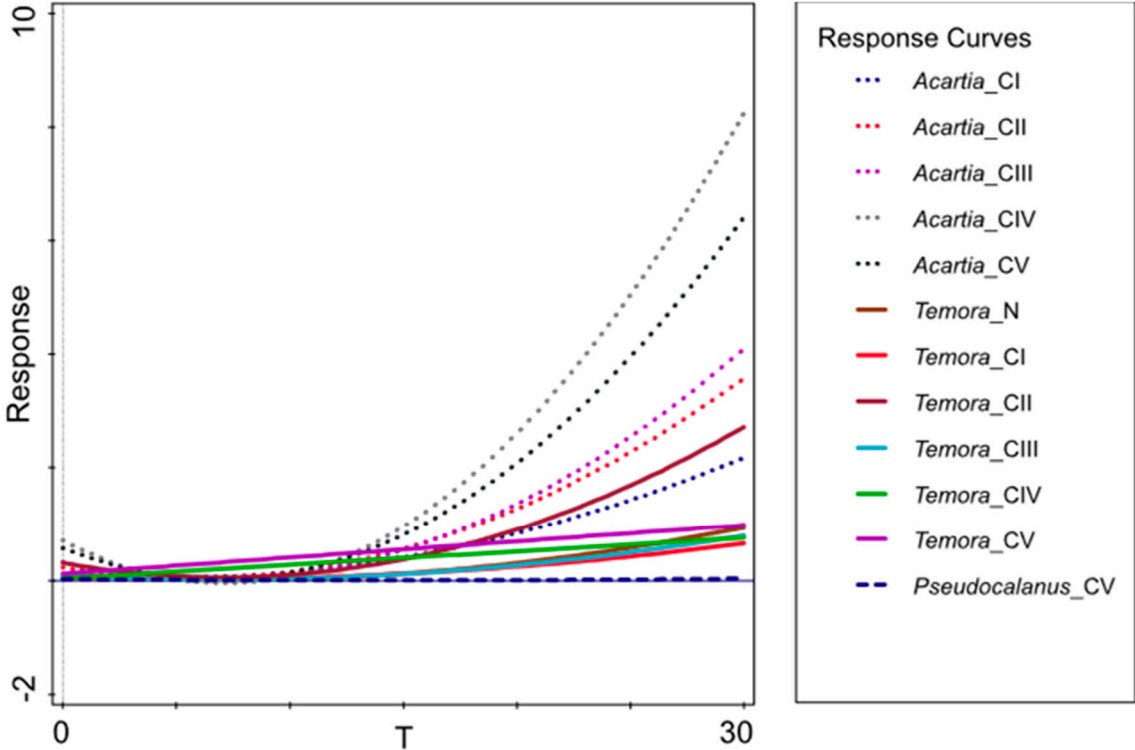

**Figure 9.** Relationship of *Acartia* spp., *Temora longicornis*, and *Pseudocalanus acuspes* developmental stages with temperature from the generalized linear models (GLMs).

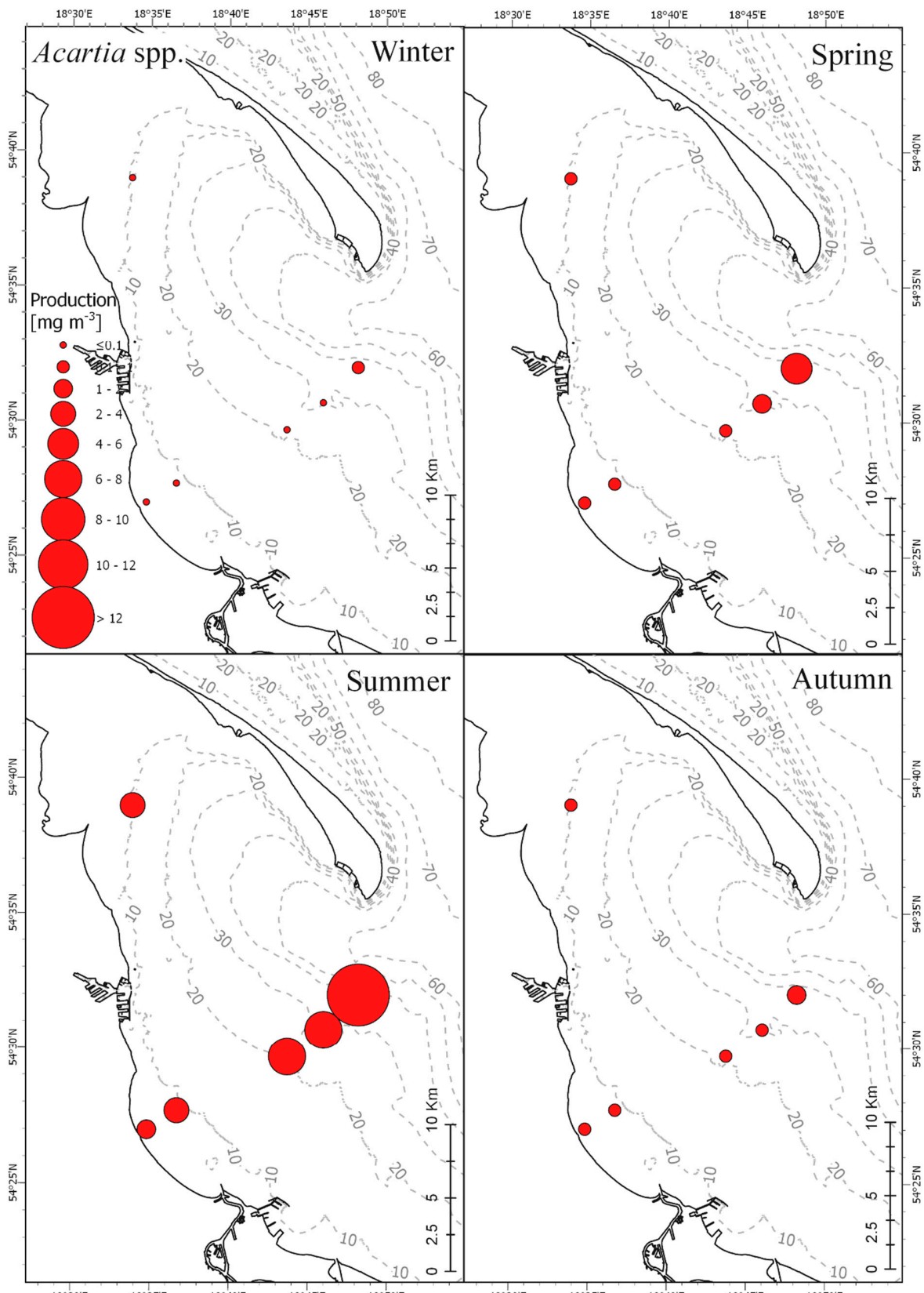

**Figure 10.** Horizontal distribution of average secondary production rates of *Acartia* spp. in the Gulf of Gdańsk.

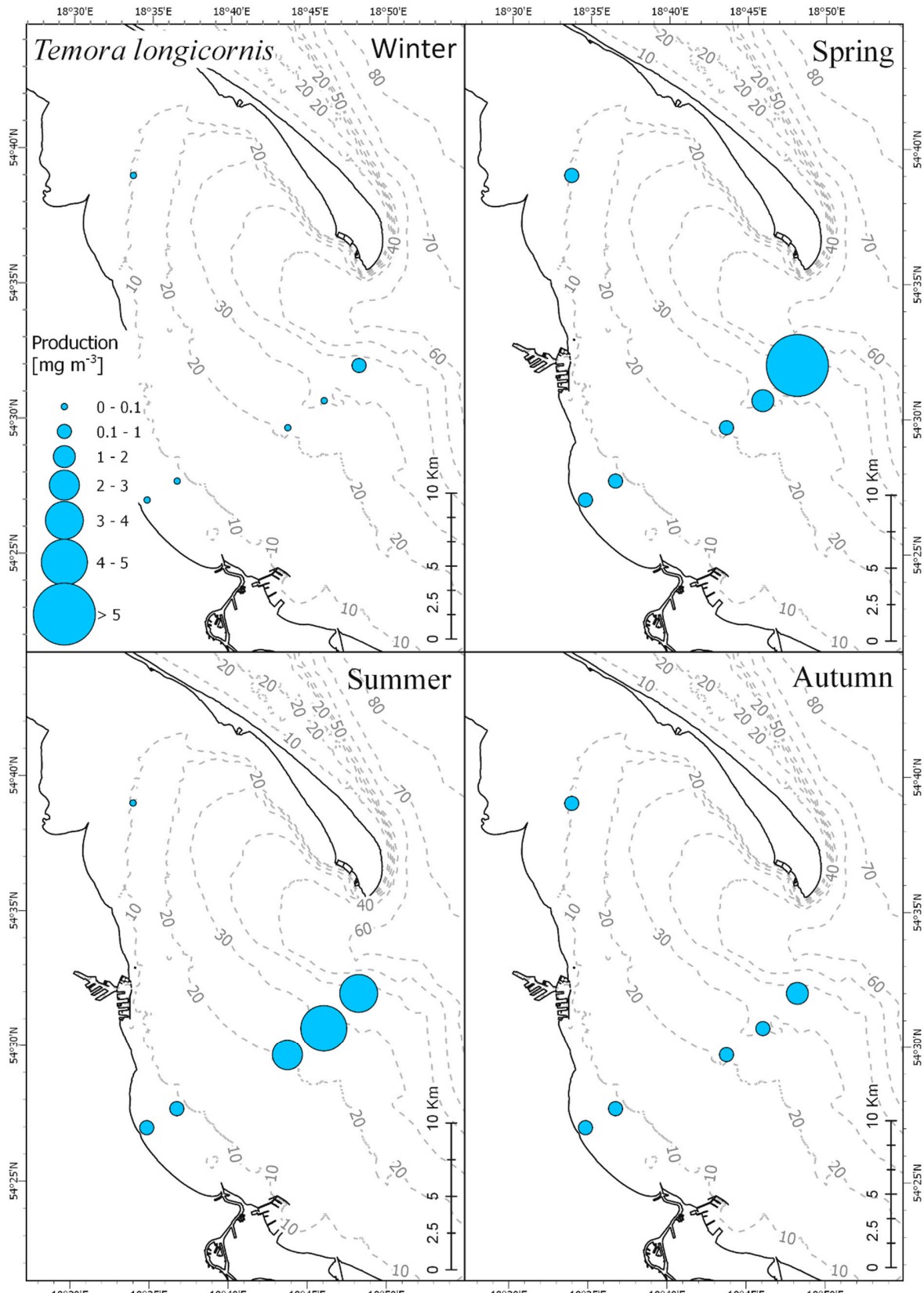

**Figure 11.** Horizontal distribution of average secondary production rates of *Temora longicornis* in the Gulf of Gdańsk.

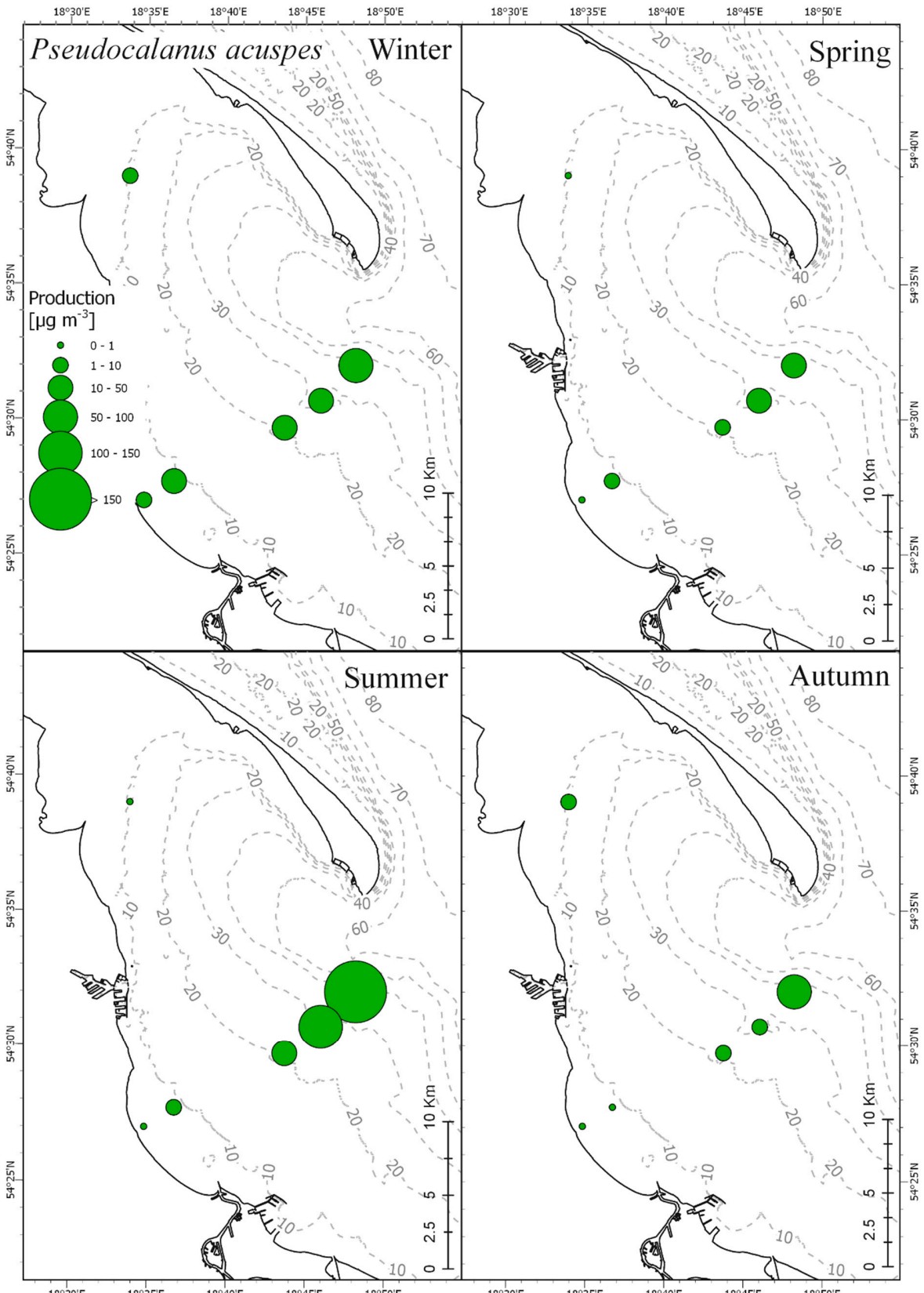

**Figure 12.** Horizontal distribution of average secondary production rates of *Pseudocalanus acuspes* in the Gulf of Gdańsk.

**Table 4.** Response of copepods developmental stages production to temperature (generalized linear model (GLM)).

| Predictors | T | | | |
|---|---|---|---|---|
| Distribution | normal | | | |
| Link function | identity | | | |
| GLM fitted for 12 response variables: | | | | |
| **Developmental stages** | **Type** | **R2[%]** | **F** | *p* |
| *Acartia* spp._CI | quadratic | 18.4 | 18.4 | <0.00001 |
| *Acartia* spp._CII | quadratic | 1.0 | 17.4 | <0.00001 |
| *Acartia* spp._CIII | quadratic | 20.5 | 21.0 | <0.00001 |
| *Acartia* spp._CIV | quadratic | 24.6 | 26.6 | <0.00001 |
| *Acartia* spp._CV | quadratic | 20.9 | 21.5 | <0.00001 |
| *Temora longicornis*_N | quadratic | 19.5 | 19.8 | <0.00001 |
| *Temora longicornis*_CI | quadratic | 12.2 | 11.3 | 0.00002 |
| *Temora longicornis*_CII | quadratic | 15.0 | 14.3 | <0.00001 |
| *Temora longicornis*_CIII | quadratic | 14.0 | 13.3 | <0.00001 |
| *Temora longicornis*_CIV | linear | 5.5 | 9.6 | 0.00229 |
| *Temora longicornis*_CV | linear | 4.3 | 7.4 | 0.00732 |
| *Pseudocalanus acuspes*_CV | quadratic | 8.3 | 7.4 | 0.00086 |

## *3.4. Mortality Rate*

Mortality rates of the investigated species were diverse thorough the research period. The result obtained for *Acartia* spp. showed the highest mortality rates mainly in summer, although in the initial years of research a different trend was observed. In 1998, the highest mortality rate was recorded in autumn (0.4 day$^{-1}$); while in 2000 and 2006 the peak mortality rate was recorded in spring. The highest rate of mortality of *Acartia* spp. was obtained in summer 2010, with a value of 0.8 day$^{-1}$. Winter was characterized by lowest mortality, remaining during all years of research at a similar level: about 0.2 day$^{-1}$. Increased mortality rates of *T. longicornis* were usually noted for two seasons: summer–autumn 1999, spring–summer 2007, and spring–summer 2010–2012. These values fluctuated between 0.5 day$^{-1}$ to 0.7 day$^{-1}$. However, the highest mortality rate of *T. longicornis* was observed in spring 2006, reaching over 0.8 day$^{-1}$. In winter mortality remained low, not exceeding 0.2 day$^{-1}$. The mortality rates of *P. acuspes* showed a very irregular distribution throughout the study period. In 1999 and 2007, two peaks were recorded, in winter and in summer. During 2006 and 2010 increased mortality of this species was observed during the summer season. Mortality rates in 2011 and 2012 remained at a similar level in all seasons (Figure 13).

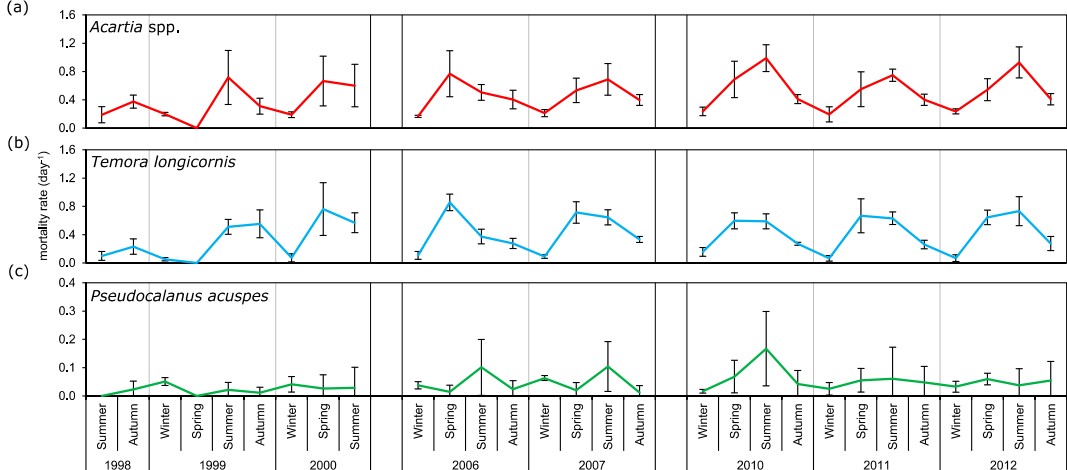

**Figure 13.** Mean daily mortality rates (± standard deviation) for (**a**) *Acartia* spp., (**b**) *Temora longicornis*, and (**c**) *Pseudocalanus acuspes* in the Gulf of Gdańsk during seasons of investigation.

In context of mortality of individual development stages for *Acartia* spp. (N-CV) the highest mortality was noted in the spring–summer period, and it concerned mainly the fifth copepodite stage (CV). Maximum values were noted in spring 2006 and 2010—~0.3 day$^{-1}$—while in spring 2000 a high mortality rate was noted for CIII. In the second research period there was an increase in mortality of nauplii, persisting from summer 2006 to winter 2007 (oscillating around 0.1 day$^{-1}$); in summer 2007 a high mortality rate of CIV (0.24 day$^{-1}$) was also observed. The third time interval was characterized by high mortality, mainly among nauplii—~0.2 day$^{-1}$—and CV mortality, between spring and autumn 2011 and 2012 (Figure 14a).

Considering mortality rates of *T. longicornis* stages during the first study period, the highest values were observed for CIII, CIV, and CV, ranging from 0.15 day$^{-1}$ to 0.28 day$^{-1}$. The lowest mortality rate during that period was recorded for nauplii. In 2006–2007 and 2010–2012, the highest mortality was observed for stages CII, CIII, and CIV in spring and for nauplii in the summer, while for CV two mortality peaks were noted, the first in spring and the second one in the autumn (Figure 14b).

For *P. acuspes*, chaotic distribution of mortality of all stages was noted, which may have been caused by a relatively low abundance of that species. High mortality rates for CIII in winter of 2010 and 2012 and for CIV in winter 2011 were noticeable (Figure 14c).

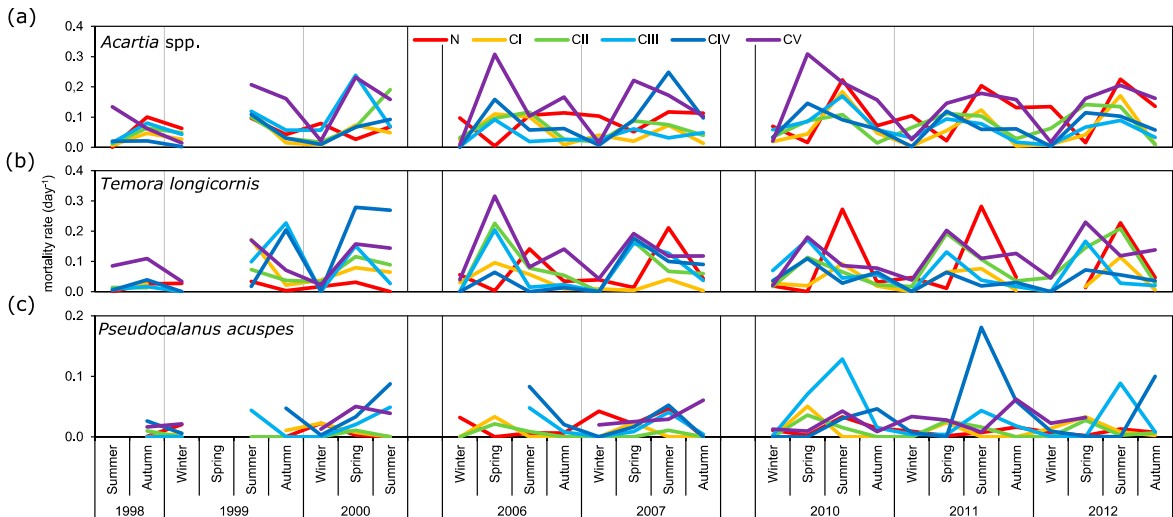

**Figure 14.** Daily mortality rates of copepod stages N–CV for (**a**) *Acartia* spp., (**b**) *Temora longicornis*, and (**c**) *Pseudocalanus acuspes* in the Gulf of Gdańsk during seasons of investigation.

When considering mortality rates in relation to seasons, significant differences were observed ($p = 0.001$, global R = 0.355). The most visible differences in mortality rates were noted between winter and spring seasons ($p = 0.001$, global R = 0.584), as well as between autumn and winter ($p = 0.001$, global R = 0.539) (Figure 15).

Mortality of *Acartia* spp. and *T. longicornis* was the lowest in autumn and winter for almost all stages, with the exception of nauplii of both species, which achieved the lowest values of mortality during spring. *P. acuspes* mortality distribution, as in the case of production analysis, was unspecified. However, nauplii demonstrated the highest mortality in summer and winter, but CI and CII in spring. CIII and CIV, in turn, indicated the highest mortality in summer. *P. acuspes* CV presented almost equal mortality for all seasons, but during autumn the range of values was the widest (Figure 16).

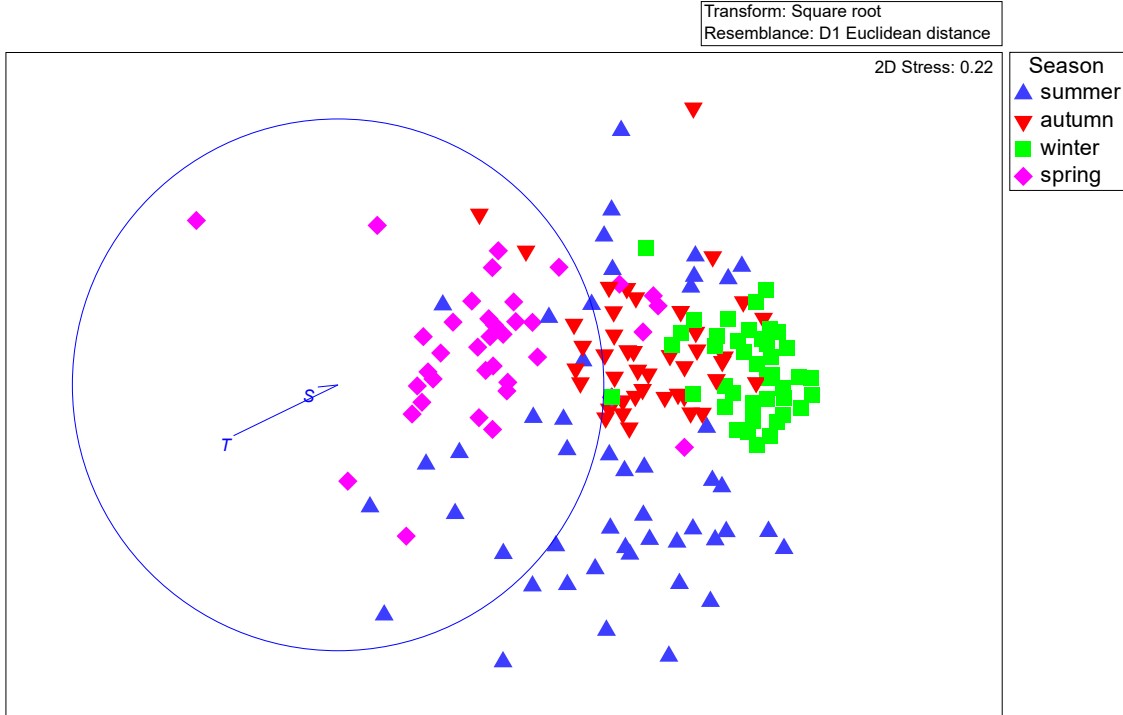

**Figure 15.** nMDS plot illustrating the samples ordination according to mortality rates based on the seasons factor.

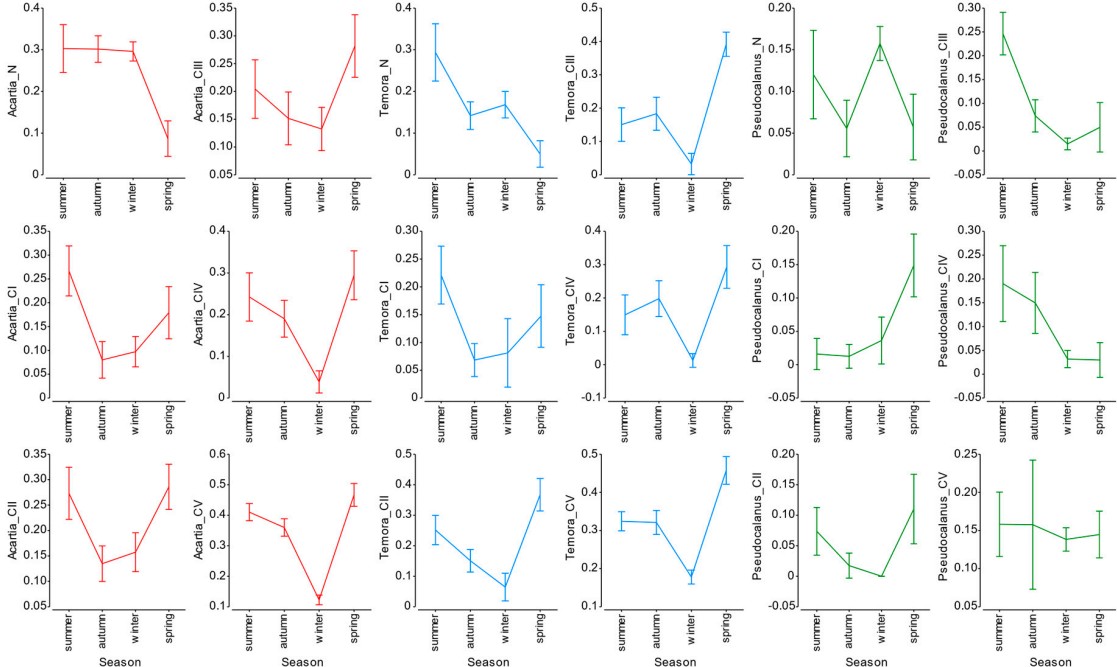

**Figure 16.** Means plot illustrating mortality distribution of *Acartia* spp., *Temora longicornis*, and *Pseudocalanus acuspes* developmental stages in relation to seasons.

Considering mortality rates, GLM plots showed a positive, linear relationship between *T. longicornis* stages (from N to CIV), *Acartia* spp. (CI, CII, and CIV), and temperature. Both *Acartia* spp. CV and *T. longicornis* CV demonstrated unimodal response along the temperature gradient, while *P. acuspes* CV showed a negative relationship with temperature factor (Table 5, Figure 17).

**Table 5.** Response of copepods developmental stages mortality to temperature (GLM).

| Predictors | T | | | |
|---|---|---|---|---|
| Distribution | normal | | | |
| Link function | identity | | | |
| GLM fitted for 11 response variables: | | | | |
| **Developmental stages** | **Type** | **R2[%]** | **F** | ***p*** |
| *Acartia* spp._CI | linear | 5.1 | 8.9 | 0.00331 |
| *Acartia* spp._CII | linear | 4.1 | 7.1 | 0.00868 |
| *Acartia* spp._CIV | linear | 11.5 | 21.3 | <0.00001 |
| *Acartia* spp._CV | quadratic | 20.3 | 20.7 | <0.00001 |
| *Temora longicornis*_N | linear | 5.4 | 9.3 | 0.00266 |
| *Temora longicornis*_CI | linear | 8.6 | 15.5 | 0.00012 |
| *Temora longicornis*_CII | linear | 5.6 | 9.8 | 0.00206 |
| *Temora longicornis*_CIII | linear | 4.5 | 7.7 | 0.00617 |
| *Temora longicornis*_CIV | linear | 4.0 | 6.9 | 0.00946 |
| *Temora longicornis*_CV | quadratic | 8.6 | 7.7 | 0.00064 |
| *Pseudocalanus acuspes*_CV | linear | 4.8 | 8.2 | 0.00478 |

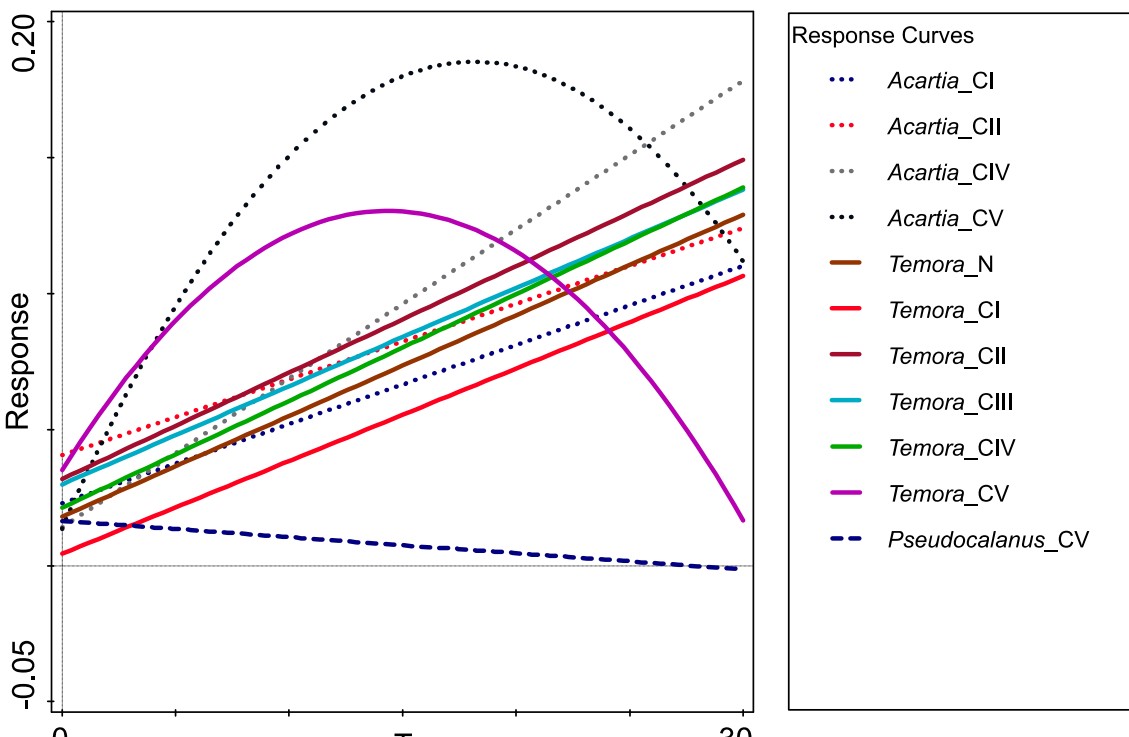

**Figure 17.** Relationship of *Acartia* spp., *Temora longicornis*, and *Pseudocalanus acuspes* developmental stages with temperature from the generalized linear models (GLM).

Horizontal mortality distribution of *Acartia* spp. showed the highest mortality rate at the shallow station 6 in spring and at stations 6, 1, and 2 (> 0.68) in summer. In autumn, the mortality rate of *Acartia* spp. was within the same range at all stations (0.34–0.68). The lowest mortality was recorded at station 6 in winter (0–0.17) (Figure 18).

The results for *T. longicornis* showed the highest mortality rates at stations 6 and 2, as well as at station 3 in spring (> 0.64). In the summer season, at all stations, the mortality rate was in the same range (0.32–0.64). The lowest values were recorded in the winter at stations 6, 4, and 5 (0–0.08) (Figure 19).

For *P. acuspes*, the horizontal mortality distribution showed the highest mortality at deep stations 3 and 4 in the summer season (>0.08). However, the lowest mortality was recorded at shallow stations 1 and 6, also in summer, and at stations 1 and 2 in autumn (Figure 20).

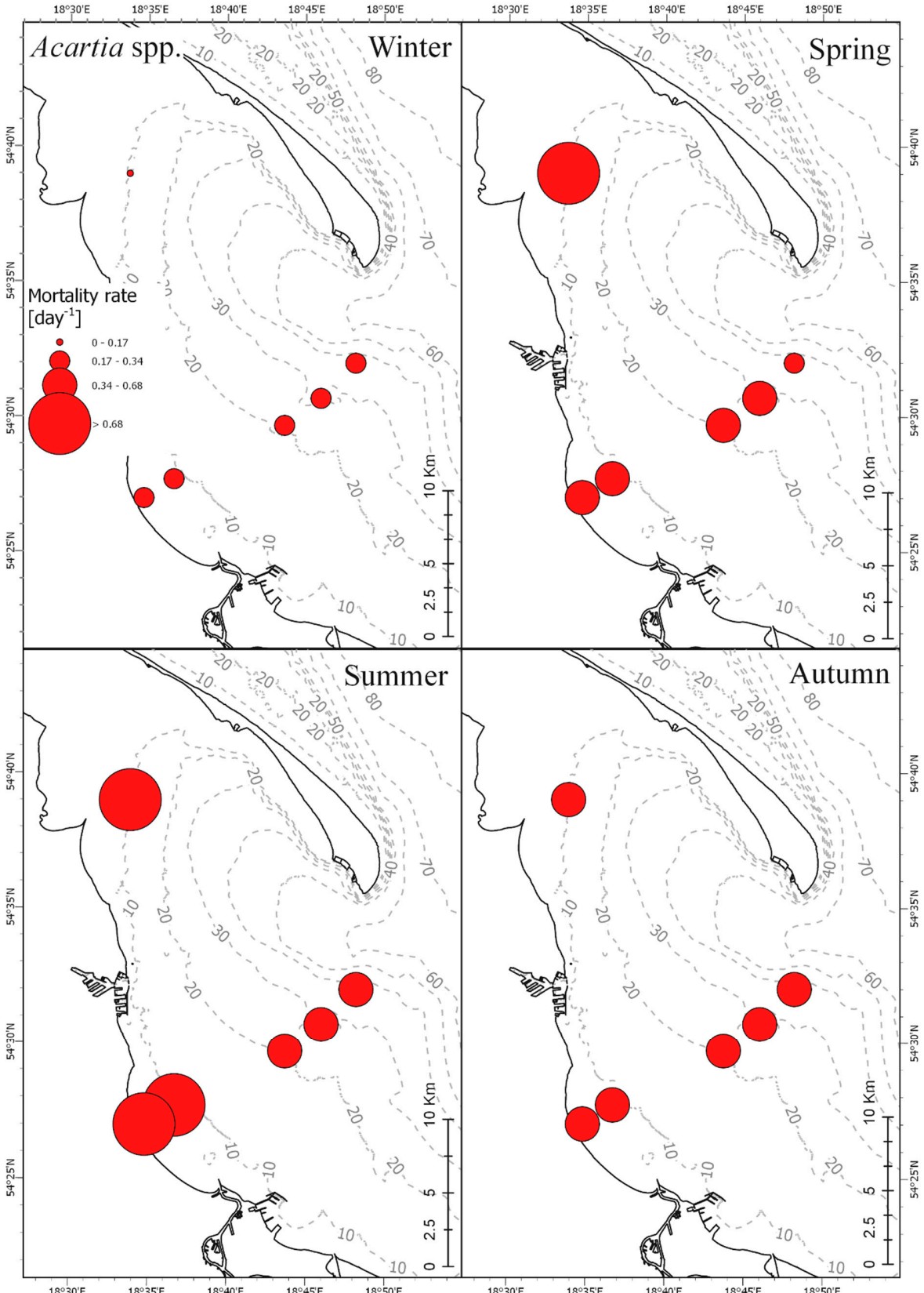

**Figure 18.** Horizontal distribution of average mortality rates of *Acartia* spp. in the Gulf of Gdańsk.

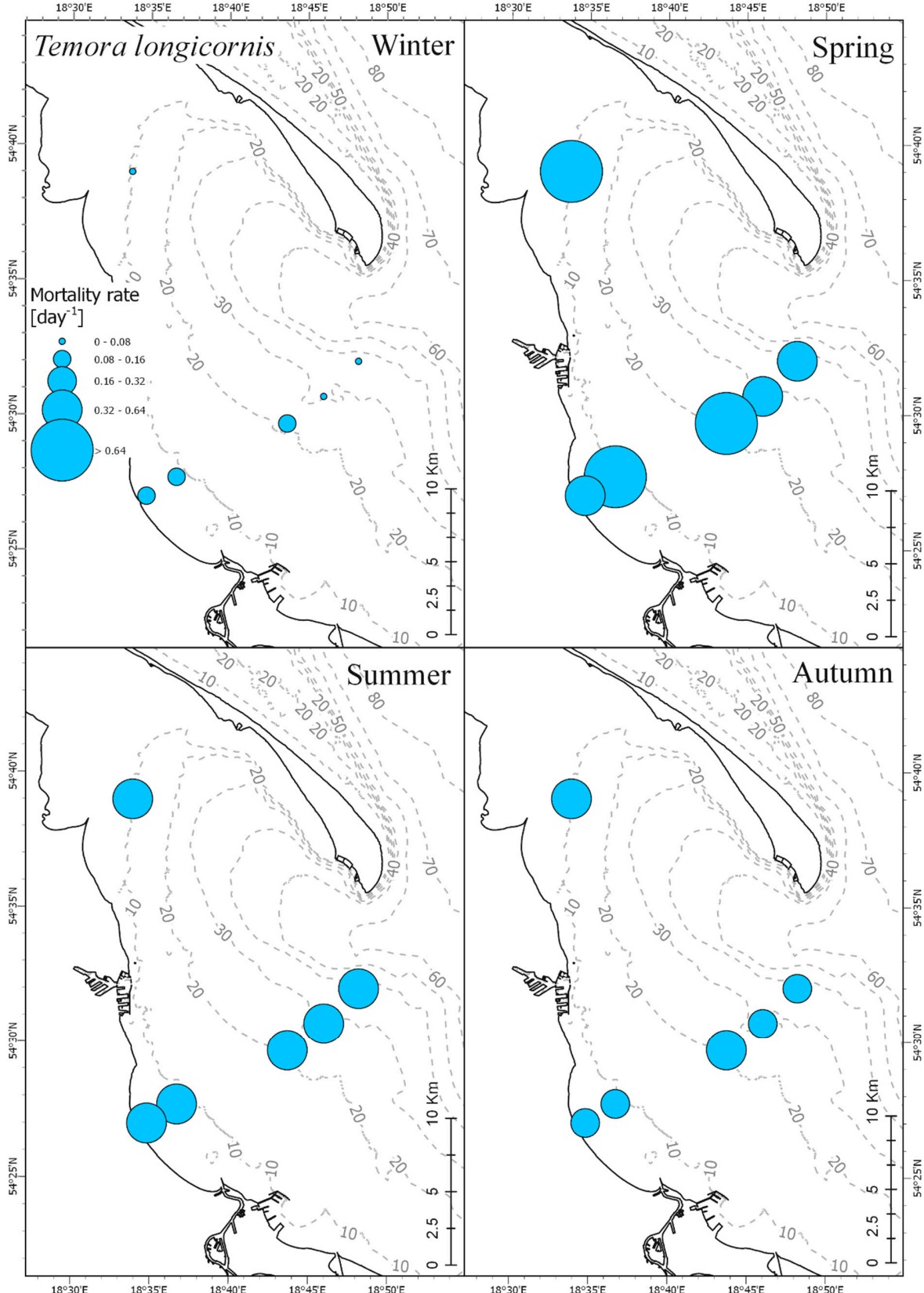

**Figure 19.** Horizontal distribution of average mortality rates of *Temora longicornis* in the Gulf of Gdańsk.

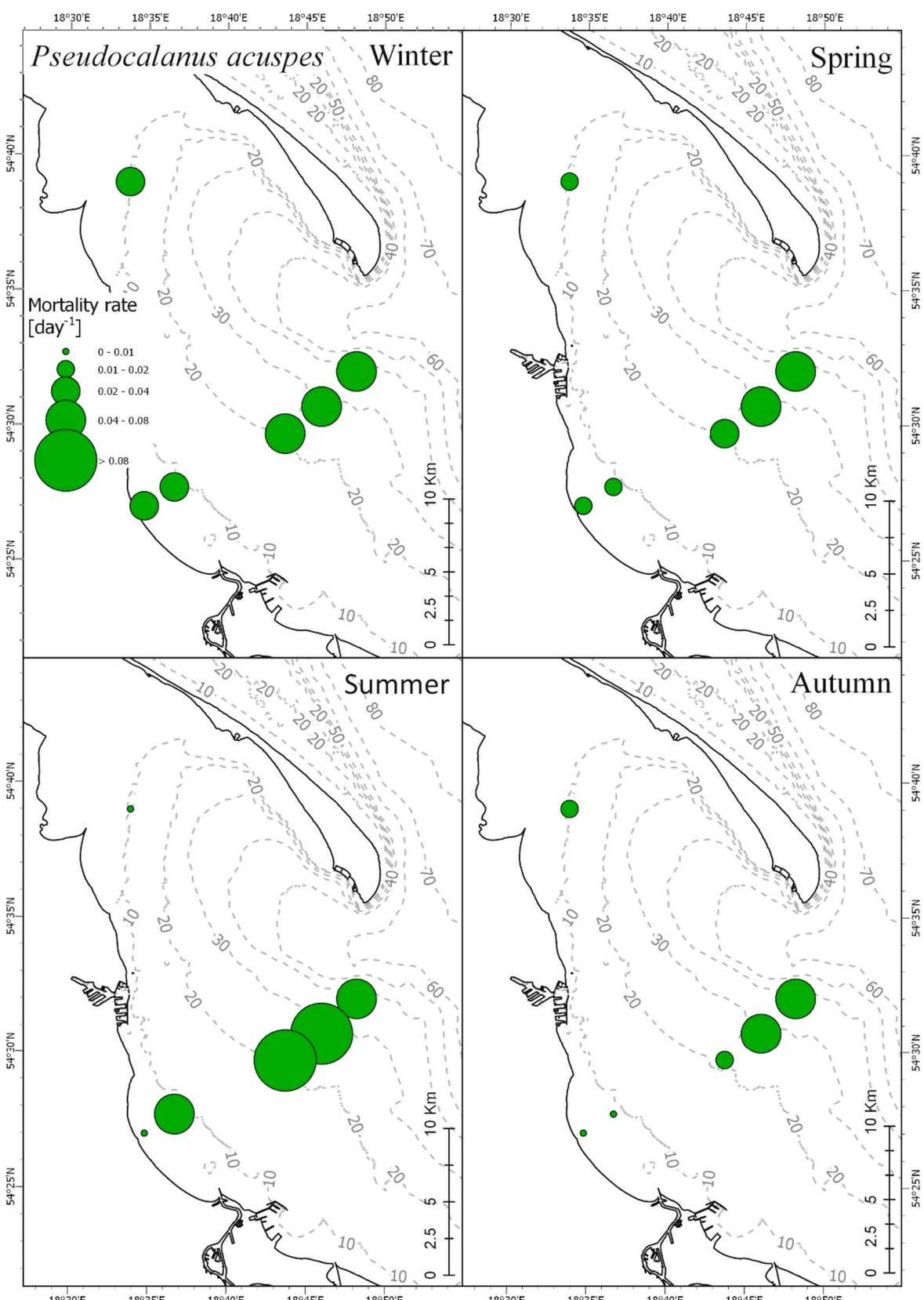

**Figure 20.** Horizontal distribution of average mortality rates of *Pseudocalanus acuspes* in the gulf of Gdańsk.

## 4. Discussion

The aim of our study was a description of seasonal and interannual patterns of secondary production and mortality rates for the main southern Baltic copepods. The main factors determining zooplankton production are temperature and food availability [29]. Therefore, we decided to investigate seasonal and interannual fluctuations of secondary production of three copepod taxa: *Acartia* spp., *T. longicornis*, and *P. acuspes* in the southern Baltic. This is even more important in the context of increasing water temperatures observed in the Baltic related to global warming [30]. Higher water temperature leads to shorter generation time and smaller body size of copepods, also causing individuals to reach reproductive age quicker, and causing rapid increases in density [31]. However, different copepod species have their individual temperature optimums, at which their development is most optimal [10]. Therefore, when estimating the rate of secondary production of these copepods, we used a *Di* function that takes into account the individual temperature optimums. We are also aware that accurate estimates, most approximate to the natural state of the secondary production based on mathematical expressions, require a method that combines a variety of factors. Because of that our estimations do not fully reflect actual, real as in the natural environment, secondary production values. However, they allow for an approximation of these values and enable recognition of trends or anomalies occurring in the studied ecosystem.

Results obtained in our research showed clear correlation between seasonal production fluctuations in the Gulf of Gdańsk and the hydrological conditions, mainly water temperature (Figure 21). The highest correlation was recorded during summer, mainly for the young copepodite stages of *Acartia* spp. as well as nauplii and copepodites of *T. longicornis*. This is consistent with research carried out by Koski et al. [32] in the North Sea, which also indicates that the production coefficient is significantly positively correlated with the average water temperature. However, research from the Western Scheldt Estuary [33], showed that neither the biomass nor the secondary copepod production was associated with chlorophyll concentration, and the temperature seemed to have a significant impact only on the predominance of certain copepods. In contrary to those two taxa, *P. acuspes* showed mostly negative correlation of secondary production with temperature. This species in the central Baltic is associated with the deeper, more saline, colder water layer [34–36]. This was visible in high mean values of secondary production of nauplii and older copepodites (CIV and CV) noted during winter seasons (Figure 8). We can therefore clearly state that, similarly to other water basins, water temperature is one of the main factors controlling not only biomass and abundance [17,37] but also secondary production of main copepod taxa in the Gulf of Gdańsk. Temperature was responsible for 11.8% of variability observed in RDA. In addition to temperature, variability of secondary production was also to some extent explained by the concentration of dissolved oxygen (5%) and chlorophyll *a* (4.5%) (Table 6).

Our results indicate that the water salinity does not have a statistically significant effect on abundance of copepodite developmental stages (Table 6). There was also no correlation for both production and mortality rates. These results differed from literature data from other brackish [38] or low salinity water bodies [36,39]. For example, in the Lake Waihola both elevated water temperature and salinity affected population dynamics of *Boeckella hamata*. Additionally, higher temperature and salinity, which were related to the increase of egg production, were offset by the higher mortality rates of reproductive females [38]. Nagaraj, on the other hand, described that the combination of temperature and salinity had a greater impact on mortality of *Eurytemor affinis* than each of these factors separately [40]. In the German Southern Bay (North Sea) [36], due to the high summer temperature, higher salinity, and sufficient food concentration, *Pseudocalanus elongatus* have three to four generations per year; similarly in the northern part of the North Sea three generations are usually described [41]. While Evans [42] described four to six generations of that species in coastal waters of Northumberland (also North Sea). However, in the Southern Baltic, due to lower salinity only one full generation per year for *P. acuspes* is observed [43]. Such a discrepancy of results is a result of the females' physiological abilities, due to slower growth and differences in the body size which are related to lower salinity. This is a well-known phenomenon concerning many organisms from the Baltic Sea, including copepods;

*T. longicornis* and *P. acuspes* occurring in the North Sea reach significantly larger body sizes than those recorded in the Baltic Sea.

**Table 6.** Conditional term effects of particular environmental variables for redundancy analysis (RDA). Explained variability and *p*-values of particular environmental variables.

| Name | Explains % | pseudo-F | *p* |
|---|---|---|---|
| Temperature (T) | 11.8 | 15.6 | 0.001 |
| Dissolved oxygen ($O_2$) | 5.0 | 7.0 | 0.002 |
| Chlorophyll *a* | 4.5 | 6.6 | 0.005 |
| Salinity (S) | 0.5 | 0.8 | 0.442 |

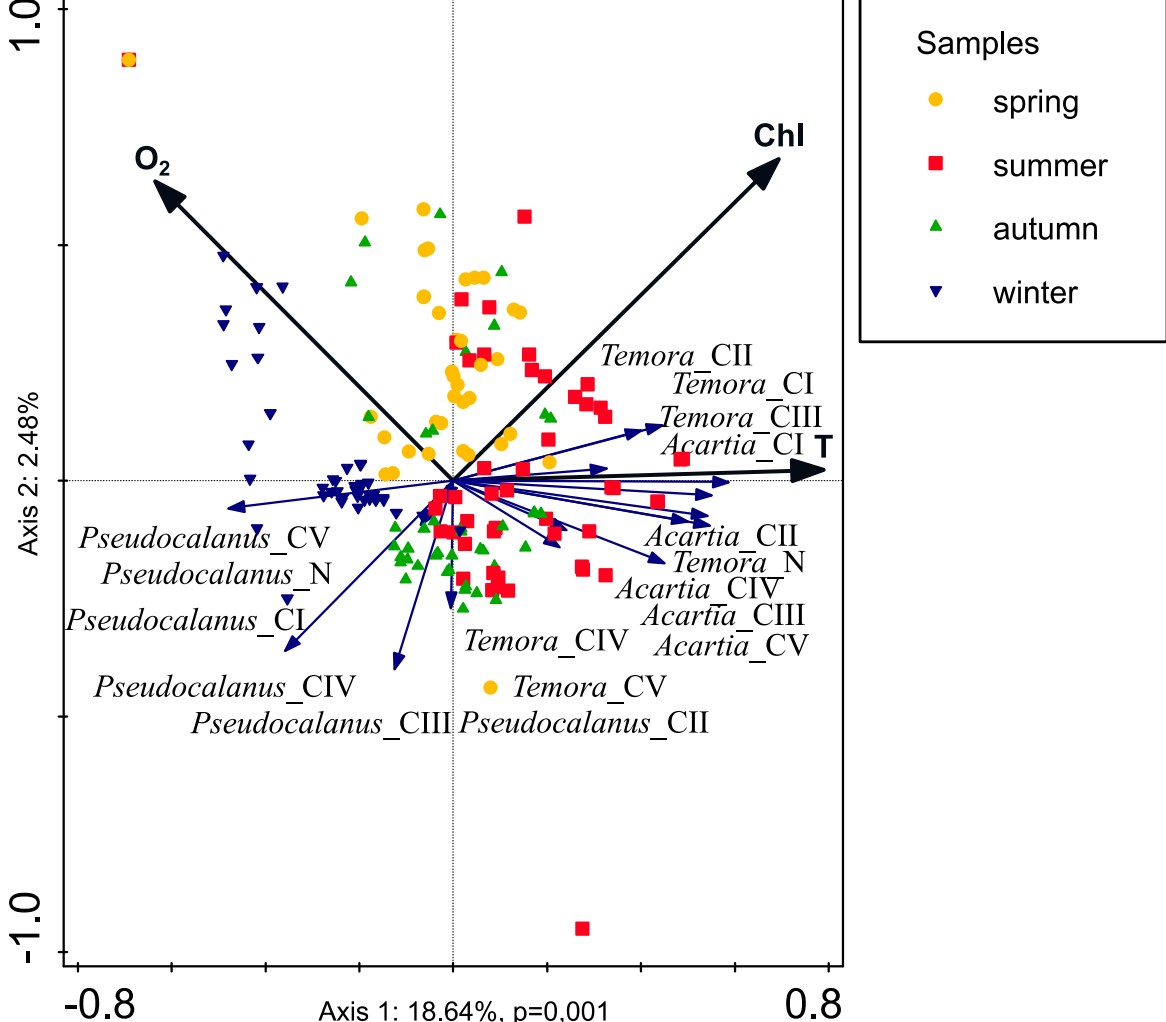

**Figure 21.** Ordination plot from redundancy analysis (RDA) on secondary production of development stages of *Acartia* spp., *Temora longicornis*, and *Pseudocalanus acuspes* (N—nauplii; CI–CV—copepodites of respective stage) and their relation to chlorophyll *a* and dissolved oxygen.

Comparison of the estimated values of secondary production of crustaceans from other regions shows that that copepod production in the Gulf of Gdańsk is relatively low. The maximum estimated average value of *P. acuspes* was 0.2 mgC m$^{-2}$ day$^{-1}$ in the summer of 2007, while Renz [44] described it as 26.2 mg C m$^{-2}$ day$^{-1}$ in June. Such a large difference in the obtained values is probably due to differences in hydrological factors leading to differences in metabolic rates [36]. Renz and Hirche [36] have shown that the rate of North Sea *P. elongatus* population development is ~3–5 times higher and the growth rate is up to 10 times higher than the *P. acuspes* population from the Baltic Sea, which translates

directly into the higher production of *P. elongatus*. Results reported by Fransz [45] from the North Sea show that the secondary production of *T. longicornis* and *Acartia clausi* fluctuated between 22 and 16 mgC m$^{-2}$ between May and September. In the brackish waters of the Western Scheldt Estuary, the estimated maximum average secondary production of *Acartia tonsa* oscillated around 25 mgC m$^{-3}$ in August; the second peak was recorded in September with the value of 8 mgC m$^{-3}$. The presence of maxima of secondary production in these months is consistent with the seasonality of production observed for these species from genus Acartia in the Gulf of Gdańsk. Production values, as in the case of *P. acuspes*, were, however, lower than those recorded in the North Sea. Differences in the value of secondary production can also be caused by the use of different calculation methods and wet mass converters for the copepod from various sources of literature.

In the perspective of the observed progressing warming of the Baltic Sea, which is particularly noticeable in its northern region (the air temperature in the spring increased by ~1.5 °C over the period of 1871 to 2011 [46]. The ecosystem of the southern Baltic, which is much more productive and biodiverse, is more susceptible to the negative effects of such changes. Due to the currently observed restructuring of unicellular plankton and the shift of phenological phases [47], further comprehensive research on biological production in ecosystems is needed, combining the research on primary phytoplankton production and the production of zooplankton. Observed progressing delaying of the maximum of secondary production in relation to spring bloom of phytoplankton may lead to serious consequences for the whole organic production and higher trophic levels in the ecosystem.

Zooplankton mortality estimates are still not a well-developed parameter in determining population dynamics. Therefore, selecting a proper methodology to describe this phenomenon can be challenging, and the main uncertainty results from inherent difficulties in measuring this process in the natural environment [48].

It is widely accepted that the daily copepod mortality decreases with consecutive developmental stages or with an increase in size [49,50]. Based on these assumptions we wanted to describe mortality rates of main copepod taxa from the Gulf of Gdańsk at particular seasons of the year. The results obtained for *Acartia* spp. and *T. longicornis*, however, show differences in mortality for particular stages from that described by the above-mentioned authors. The highest mortality rate for *Acartia* spp. was observed for the oldest copepodites (CV) with maximum values noted in spring: ~0.3 day$^{-1}$. For *T. longicornis*, high rates were observed for CIII, CIV, and CV, within a range of 0.15 day$^{-1}$ to 0.28 day$^{-1}$. High variability in mortality estimates between copepod species and developmental stages in both spatial and seasonal distribution were also observed by other authors [15,51,52].

In our research we observed cyclical changes in mortality, with the peak falling in the spring and summer season. Maud et al. [53], on the other hand, recorded the highest mortality rates in summer and autumn, with the lowest—as in our research—in winter. Differences in seasonality of mortality in different regions may result from differences in the main cause of mortality. Mortality of copepods can be caused by predatory (consumptive) or physicochemical and biological factors as factors causing nonconsumptive mortality. Consumptive mortality may be associated with the abundance of predators [54], and is described in the literature as usually occurring in the autumn season, when the abundance of predators is the highest. This type of mortality described for *Calanus helgolandicus* constituted an average of 89% of the total mortality for this species [53]. In the Baltic Sea, copepods are a valuable source of food for the commercially important fish species sprat and herring, but jellyfish can also have a significant predatory impact. The gelatinous zooplankton was the main cause of mortality variability observed in deep coastal sampling station located near the southwest of Plymouth, UK [55]. Nonconsumer mortality may result from death caused by age [56], diseases, and parasitism [57], exposure to environmental pollution [58], and physiological stress [59]. Field and laboratory studies show that nonconsumptive factors can account for 25–33% of the total death rate among adult copepods [60].

The species from genus Acartia had the highest abundance and biomass among the three investigated taxa, while *P. acuspes* was far less abundant than the other two taxa. Such proportions

between these taxa are quite typical for the coastal region of the southern Baltic. *P. acuspes* tends to dominate offshore areas of the Baltic Sea, is much less abundant in the coastal zones, and it is rarely present above the thermocline, especially during the warm season [61,62].

Observed long-term biomass means for *Acartia* spp. were in a similar range as those reported for this region by Möllmann [30], while for *T. longicornis*, and especially *P. acuspes*, they were much lower. This was probably due to our sampling stations being located mostly in the inner, coastal part of the Gulf of Gdańsk. Möllmann [30] also showed mostly negative anomalies for both *T. longicornis* and *P. acuspes* as well as the mostly neutral anomaly of *Acartia* spp. during the late 1990s. This is consistent with our findings, which also showed strong positive anomalies for those taxa during the first decade of the 2000s.

Obtained data show seasonal fluctuations of copepod abundance, biomass, production, and mortality rates, as well the correlation with environmental variables influencing their development and mortality. The results of this work will be useful for future extended evaluation of copepod production with relation to environmental variables in brackish waters.

**Author Contributions:** Conceptualization, L.D.-G. and M.M.-K.; funding acquisition, L.D.-G.; investigation, M.M.-K., M.K. and A.L.; methodology, M.M.-K., A.L, P.P. and M.J.; project administration, L.D.-G.; and supervision, M.I.Ż.; data visualization, M.K. and P.P.; writing—original draft, M.M.-K., M.K., A.L., P.P. and M.J.; writing—review & editing, L.D.-G.; all authors agreed to submission of the manuscript.

**Funding:** Partial support for this study was provided by the project 'Integrated info-prediction Web Service WaterPUCK', no. BIOSTRATEG3/343927/3/NCBR/2017.

**Acknowledgments:** This study has been conducted using E.U. Copernicus Marine Service Information. Calculations were carried out at the Academic Computer Centre in Gdańsk.

**Conflicts of Interest:** The authors declare no conflict of interest.

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
