# Peer review of "The Interannual Changes in the Secondary Production and Mortality Rate of Main Copepod Species in the Gulf of Gdańsk (The Southern Baltic Sea)"

_applsci, doi:10.3390/app9102039_

Round 1

Reviewer 1 Report

The draft presented work on the interannual variation of secondary production and mortality rate of three copepod species. The effect of temperature, dissolved oxygen, chlorophyll, and salinity on the biomass abundance and the mortality were discussed. The authors did solid work and the data was properly discussed. Here are some suggestion to improve the draft.

The authors may want to decrease the number of figures and tables. 

Lines 181 and 182. What is the depth of the sampling?

Figure 5. It is kind of hard to read the date. Suggest the authors just list the month here. 

Figure 6. The standard deviation is high for the summer section. How did the authors evaluate the accuracy of the data and draw a conclusion based on the data?

Lines 406 and 407. Can the authors evaluate the difference between the approximation and the actual values? How can the authors conclude the approximation is able to reflect the treads? 

Table 6. The conditions have inner correlations. How did the authors identify that on the variability in RDA?

Lines 448-450. The authors cited previous work and draw a conclusion based on the comparison of data. Are the same calculation methods used for the literature and this work?

Author Response

Dear Reviewer,

Thank you for your valuable comments, all of them will surely be helpful in improving the manuscript.

As for suggestions:

Lines 181 and 182. What is the depth of the sampling?

Data of dissolved oxygen concentrations were obtained from the Copernicus model. Data were derived for the locations of in situ sampling stations with the method described in chapter 2.3. Model data. Maximum depth of the sampling stations is visible on the map – Figure 1 (station 1 – 5 m, 2 – 10 m, 3 – 20 m, 4 – 30 m, 5 – 40 m, 6 – 10 m.).

Figure 5. It is kind of hard to read the date. Suggest the authors just list the month here. 

Figures have been corrected.

Figure 6. The standard deviation is high for the summer section. How did the authors evaluate the accuracy of the data and draw a conclusion based on the data?

High deviation is a result of large spatial variability of zooplankton, probably as an effect of patchiness. Additionally, summer is a period when Gulf of Gdańsk became thermally stratified, resulting in large differences in vertical distribution of zooplankton also affecting standard deviation values. This is a normal occurrence and should to be expected, samples were analyzed according to procedure which is wildly accepted and used in monitoring of zooplankton in the Baltic Sea.

Lines 406 and 407. Can the authors evaluate the difference between the approximation and the actual values? How can the authors conclude the approximation is able to reflect the treads? 

The aim of our research was to estimate the value of Copepoda's secondary production in the Gulf of Gdańsk, using the mathematical methods described in the literature, which are also commonly used for other water bodies. Secondary production is tricky to measure in situ and we did not have such possibility. We assume, however, that described methodology have been validated to the point where it can be used with enough level of trust, thus allowing as to approximate trends in the marine ecosystem.

Table 6. The conditions have inner correlations. How did the authors identify that on the variability in RDA?

Correlations showed on the redundancy analysis plot were based on tests and explorations of predictor effects. Summarized effects of explanatory variables were applied because of the strict interaction of main processes in the marine environment, hence, authors decided to applied conditional term effects of environmental factors.

Lines 448-450. The authors cited previous work and draw a conclusion based on the comparison of data. Are the same calculation methods used for the literature and this work?

The method in our research differs from that described by Renz (2007) for calculating production rates of Pseudocalanus acuspes in the North Sea. However, the main point is the fact that in both methods, calculations are based on the single stage growth rate and its mass, therefore they are comparable.

Reviewer 2 Report

The work deals with interesting aspects of the marine environment, in depth and providing interesting ideas for further evaluations to be extended in other realities; should be deepened links with other realities especially studied on the parameters that influence this development and mortality. In particular, it is not clear how salinity influences development and in figure 8 first graph the absence of production is not clear or should be motivated more in the discussion.

Some parts should be summarized for the main work purpose.

A sintetic concluding paragraph should be added.

Author Response

Dear Reviewer,

Thank you for your valuable comments, all of them will surely be helpful in improving the manuscript.

As for suggestions:

"The work deals with interesting aspects of the marine environment, in depth and providing interesting ideas for further evaluations to be extended in other realities; should be deepened links with other realities especially studied on the parameters that influence this development and mortality. In particular, it is not clear how salinity influences development and in figure 8 first graph the absence of production is not clear or should be motivated more in the discussion."

·         The analysis of salinity and developmental stages of copepods did not reveal any significant correlation for production nor mortality. Therefore, authors did not consider influence of this factor on the developmental processes to be significant, additional description was added in manuscript (Discussion, line 424)

·         Figure 8 has been corrected, plot per Acartia spp. nauplii was removed. The figure could mislead readers by showing negative output that was actually caused by a negative mass difference between nauplii and CI. Unfortunately, data for the standard weights of many copepod species in southern Baltic are highly incomplete, necessity of the future improvement in this regard is one of our main concerns.

"Some parts should be summarized for the main work purpose.

A sintetic concluding paragraph should be added."

·         Additional paragraph was added at the end of discussion chapter.